# Mechanical loading and hyperosmolarity as a daily resetting cue for skeletal circadian clocks

Michal Dudek [1,2,3], Dharshika R. J. Pathiranage[1,2,3], Beatriz Bano-Otalora [2], Anna Paszek[1,2,3], Natalie Rogers[1,2,3], Cátia F. Gonçalves[1,2,3], Craig Lawless [1,3], Dong Wang[4], Zhuojing Luo[4], Liu Yang [4], Farshid Guilak [5,6], Judith A. Hoyland [3,7] ✉ & Qing-Jun Meng [1,2,3] ✉

Daily rhythms in mammalian behaviour and physiology are generated by a multi-oscillator circadian system entrained through environmental cues (e.g. light and feeding). The presence of tissue niche-dependent physiological time cues has been proposed, allowing tissues the ability of circadian phase adjustment based on local signals. However, to date, such stimuli have remained elusive. Here we show that daily patterns of mechanical loading and associated osmotic challenge within physiological ranges reset circadian clock phase and amplitude in cartilage and intervertebral disc tissues in vivo and in tissue explant cultures. Hyperosmolarity (but not hypo-osmolarity) resets clocks in young and ageing skeletal tissues and induce genome-wide expression of rhythmic genes in cells. Mechanistically, RNAseq and biochemical analysis revealed the PLD2-mTORC2-AKT-GSK3β axis as a convergent pathway for both in vivo loading and hyperosmolarity-induced clock changes. These results reveal diurnal patterns of mechanical loading and consequent daily oscillations in osmolarity as a bona fide tissue niche-specific time cue to maintain skeletal circadian rhythms in sync.

The daily patterns of rhythmic environment on Earth, including light/darkness, temperature fluctuations and availability of food, have profound impacts on the physiology and behavior of living organisms. To anticipate and cope with changing demands between day and night, most organisms have evolved an internal cell-intrinsic timing mechanism, the circadian clock. Close alignment of internal circadian rhythms with the external environment (known as entrainment) is of paramount importance for organismal health and survival[1]. In mammals, the current model proposes that the central pacemaker—the

Suprachiasmatic Nuclei (SCN) in the hypothalamus—temporally coordinates and adjusts peripheral clocks in all major body organs[2-4]. Light as a primary and potent entrainment factor resets the central clock which in turn signals directly through neuronal connections or indirectly through hormonal cues to other parts of the brain and the body[5,6]. However, one argument for the evolution of local tissue clocks (rather than simply a top-down control from the SCN) is that it allows flexibility - peripheral clocks can adopt a different phase relationship with one another (and with the SCN) if circumstances require. If this is

[1]Wellcome Centre for Cell Matrix Research, Faculty of Biology, Medicine and Health, University of Manchester, Oxford Road, Manchester, UK. [2]Centre for Biological Timing, Faculty of Biology, Medicine and Health, University of Manchester, Oxford Road, Manchester, UK. [3]Division of Cell Matrix Biology and Regenerative Medicine, School of Biological Sciences, Faculty of Biology, Medicine and Health, University of Manchester, Manchester Academic Health Science Centre, Manchester, UK. [4]Institute of Orthopedic Surgery, Xijing Hospital, Fourth Military Medical University, Xi'an, China. [5]Department of Orthopedic Surgery and Department of Biomedical Engineering, Center of Regenerative Medicine, Washington University, St. Louis, MO 63110, USA. [6]Shriners Hospitals for Children – St. Louis, St. Louis, MO 63110, USA. [7]NIHR Manchester Biomedical Research Centre, Central Manchester Foundation Trust, Manchester Academic Health Science Centre, Manchester, UK. ✉e-mail: Judith.A.Hoyland@manchester.ac.uk; Qing-Jun.Meng@manchester.ac.uk

true, we would expect local clocks to be synchronized to the most physiologically relevant stimuli. One critical piece of evidence supporting this is the uncoupling of circadian clocks in the liver and other tissues from the SCN upon restricted feeding[7–11].

The mammalian skeletal system, especially the articular cartilage in joints and the cartilaginous tissue in intervertebral disc (IVDs) in the spine, provides a unique model to resolve this question. Cartilage and IVDs are among the most highly loaded tissues, experiencing a diurnal loading cycle associated with daily rest/activity patterns[12,13]. In humans, these tissues experience a prolonged period of compression of 2–3.5 MPa during the activity phase, followed by a period of low-load recovery during the resting phase (below 0.1 MPa). Tissues such as cartilage and IVD have high negative charges attached to the proteoglycan-rich matrix in each tissue unit/volume, which significantly affects tissue mechanical properties (including the compressive modulus) and swelling pressure. As such, cartilage and IVD tissues show profound fluctuations in their osmotic environment upon daily mechanical loading[14,15]. These loading patterns lead to diurnal changes in tissue compressive strain, resulting in corresponding changes in the osmotic environment of the cells. Such diurnal patterns may be altered due to a sedentary lifestyle, obesity or rotating shift work, contributing to the risks of developing diseases such as osteoarthritis and IVD degeneration[12,13,16,17]. Cartilage and IVDs possess intrinsic circadian clocks that drive rhythmic expression of tissue-specific genes and become dampened and out of phase in aging mice[18–21]. Genetic disruption of cartilage and IVD clocks in mice results in an imbalance of anabolic and catabolic processes which subsequently leads to accelerated tissue aging and degeneration[18–21]. The aneural and avascular nature of articular cartilage and IVDs makes them less amenable to conventional systemic time cues conveyed through nerves or hormones[22]. As such, it remains unknown how these skeletal clocks are entrained under physiological conditions. Indeed, we observed no difference in the potency of mouse serum collected during the day vs. night in their ability to synchronize these skeletal tissue clocks (Fig. S1), thus excluding blood-borne serum factors as an effective clock resetting cue for these tissues[23]. Given the diurnal loading patterns they experience, a priori should expect these peripheral skeletal clocks to be synchronized by some aspects of mechanical loading. However, until now, this has not been shown.

Here we show that daily patterns of mechanical loading and associated osmotic challenge within physiological ranges reset circadian clock phase and amplitude in cartilage and IVD tissues. Mechanistically, the PLD2-mTORC2-AKT-GSK3β axis acts as a convergent pathway for both loading and hyperosmolarity-induced clock changes.

## Results

### Mechanical loading resets the circadian clocks in femoral head cartilage and IVD tissue

We hypothesized that if mechanical loading is a key entraining factor for circadian rhythm in skeletal tissues, physical activity during the mouse resting period should shift the phase of the clock. To test this, voluntary running (e.g. on a running wheel) is not suitable as mice will not normally run during the daytime. The treadmill is the methodology of choice because it simultaneously allows for precise control of the time of exercise, intensity, and volume. PER2::Luc reporter mice were gradually adapted to daily treadmill running (45 min at 15 m/min speed) to avoid a stress response. Running occurred at ZT 2 (2 h since the resting phase), the predicted trough of *Per2* gene expression in skeletal tissues based on our RNAseq data (Fig. 1a). After 5 days of treadmill exercise, mice were sacrificed immediately after running and tissues (SCN, femoral head cartilage, and IVDs) were harvested for explant cultures. Recording of PER2::Luc bioluminescence from the tissues showed no effect of such exercise bouts on the circadian rhythm of the SCN. However, the circadian phase of the clocks in cartilage and IVD tissues was advanced by ~8 h, indicating a decoupling

effect of physical exercise between the skeletal clocks and the central SCN clock (Fig. 1b, c). To exclude the potential metabolic or systemic effects of exercise on skeletal clocks, we tested directly the role of mechanical loading using an ex vivo PER2::Luc tissue explant culture model and the FlexCell compression system. A short loading regime (1 h of 1 Hz, 0.5 MPa compression) resulted in a robust amplitude increase of the circadian rhythm in cartilage which was maintained for at least three more days (Fig. 1d). To delineate the best response window within the 24-h cycle, we applied the same compression protocol at 4 time points 6 h apart. Only compression applied at the peak of PER2::Luc resulted in a significant increase of circadian amplitude, with a minimal phase shift. Compression at other phases resulted in significant phase delay or advance ($p < 0.001$, Fig. 1e, f). Compression applied at the trough disrupted circadian rhythm, highlighting the importance of the circadian phase in modulating the clock response to loading (Fig. 1e, f). The phase shift of the cartilage circadian rhythm was dependent on the magnitude of the force of compression (Fig. 1g).

The same loading protocol altered the PER2::Luc rhythm in IVD tissues in a similar manner (Fig. 1h, I and Fig. S2a). Next, we tested whether rhythmic loading patterns can entrain circadian rhythms of endogenous clock genes in IVD cells. To this end, rat IVD cells were subjected to oppositely phased cycles of mechanical loading at 0.5 MPa of 12-h loading/12-h unloading. qPCR showed that the expression of clock genes *Bmal1* and *Cry1* were driven ~180 degrees out of phase (Fig. 1j), further supporting rhythmic loading as an endogenous clock resetting cue for cells in the skeletal system.

### Hyperosmolarity (but not hypo-osmolarity) phenocopies the effect of mechanical loading

The diurnal pattern of mechanical loading of the articular cartilage and IVDs is associated with daily changes in osmotic pressure. Under load, the pressurized interstitial fluid flows to regions of lower pressure, resulting in increased osmolarity. When unloaded, the process is reversed, causing a return to normal osmolarity. These fluctuations in osmolarity play an important role in normal skeletal physiology by promoting the synthesis of the extracellular matrix and stabilizing cellular phenotypes[24–26]. Therefore, we hypothesized that fluctuations in osmolarity associated with loading might mediate the exercise-induced clock changes. We first tested how circadian rhythms in skeletal tissues operate under different baseline osmolarity conditions. PER2::Luc IVD explants were cultured under static osmotic conditions of 230–730 mOsm for 3 days then resynchronized with dexamethasone and recorded in media at the adapted osmolarity as indicated. Here, despite the wide range of osmotic conditions, the intrinsic circadian pacemaking mechanism was still intact with equivalent circadian amplitude and phase, highlighting the ability of these skeletal tissues to adapt to static changes in their environmental osmolarity (Fig. 2a). The periodicity was maintained at close to 24 h despite drastic differences in baseline osmolarity, indicating that circadian period in these skeletal tissues is "osmolarity-compensated" (Fig. 2a).

Next, we analyzed how an acute change in osmolarity (as experienced by skeletal tissues on a daily basis) impacts skeletal clocks. Cartilage tissue explants were placed in iso-osmotic (330 mOsm) or hyperosmotic (530 mOsm) media and allowed to adapt for 3 days before synchronization and recording. When clocks gradually become desynchronized, the conditioned media were swapped between iso and hyper-adapted explants. The clock amplitude in explants that experienced an increase in osmolarity was increased to a level equivalent to the beginning of the recording, and a robust circadian rhythm continued for the next 4 days. In contrast, explants experiencing a decrease in osmolarity showed a gradual loss of rhythmicity, similar to the iso-control explants (Fig. 2b). After 4 days the media were swapped back between explants and again the tissues experiencing an increase in osmolarity showed enhanced circadian rhythm (Fig. 2b). Clocks in IVD

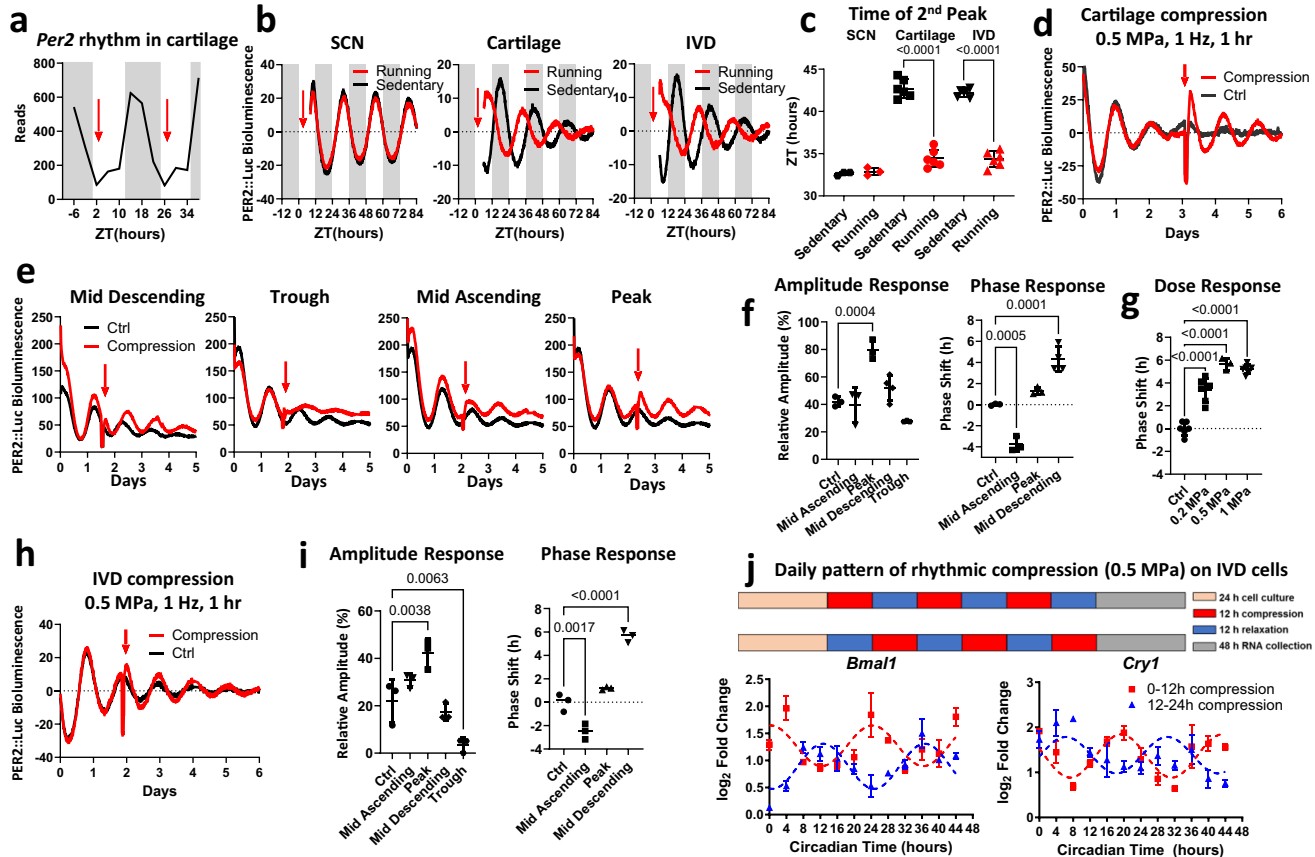

**Fig. 1 | Mechanical loading resets the circadian clock in PER2::Luc cartilage and IVDs in vivo and ex vivo. a** *Per2* mRNA expression in mouse hip cartilage from RNAseq timeseries[19]. Gray background denotes night (mouse active phase), red arrows indicate timing of treadmill running in relation to circadian phase. **b** PER2::Luc bioluminescence from mouse tissues after treadmill running (red) and sedentary control (black). Red arrow indicates time of exercise (ZT2, i.e. 2 h into resting phase). Each trace represents the mean of 3 (SCN) and 6 (cartilage, IVD) explants. **c** Quantification of phase shift between running and sedentary mice in B. **d** Bioluminescence recordings of PER2::Luc femoral head cartilage explants. Red arrow indicates time of ex vivo compression (0.5 MPa, 1 Hz, 1 h). Each trace represents the mean of 3 explants. **e** Recordings of PER2::Luc cartilage explants subjected to mechanical loading (red arrow) at 6 h intervals, starting at mid-descending phase. Each trace is the mean of 4 explants. **f** Quantification of the

PER2::Luc amplitude change in (**e**), expressed as % of the amplitude of the peak before loading and quantification of phase shifts in (**e**). **g** Quantification of phase shifts in cartilage exposed to increasing magnitude of compression applied at mid descending phase ($n = 6$ per condition). **h** Recordings of PER2::Luc IVDs subjected to compression (0.5 MPa, 1 Hz, 1 h). Each trace is the mean of 3 explants. **i** Quantification of the PER2::Luc amplitude change and phase shifts. **j** mRNA expression of *Bmal1* and *Cry1* in a rat IVD cell line following 3 cycles of oppositely phased compression (12 h at 0.5 MPa /12 h at 0 MPa). Mean ± SD of 3 cultures with 24-h cosinor curve fitting to highlight the anti-phasic nature of the gene expression profiles following the oppositely phased loading cycles. Statistical analysis was performed using one-way ANOVA. P values were adjusted for multiple comparisons using Dunnett's multiple comparisons test. Source data are provided as a Source Data file.

explants showed a very similar response to that of cartilage, with improved oscillation amplitude by increased osmolarity (Fig. S2b and Movie S1). To assess whether hyper-osmolarity synchronizes clocks at an individual cell level and to evaluate the percentage of cells that respond, single-cell fluorescence imaging of PER2::Venus mouse primary chondrocytes were performed. We observed increased nuclear PER2 signal following hyperosmotic exposure (Fig. 2c, d) and clock re-synchronization in 66% of cells (Fig. 2e).

If altered osmolarity is a key mechanism through which mechanical loading acts on the skeletal clock, it should exert very similar phase- and dose-dependent effects to mechanical loading. Indeed, a hyperosmotic challenge elicited a very similar direction of phase shifts in a circadian phase-dependent manner comparable to loading (Fig. 2f, Fig. S3). The phase response curve (PRC) and the phase transition curve (PTC) demonstrated a type 1 resetting (typical response to a relatively mild physiological clock stimulus) (Fig. S2c, d). The hyperosmolarity-induced clock-synchronizing effect appears much stronger than loading with improved circadian amplitude at all time-points except at the trough (Fig. 2g). Similar to loading, the circadian clock response to osmolarity is also "dose" dependent, with a phase

delay of up to 9.5 h (at mid-descending phase) for the +400 mOsm increase (Fig. 2h). A significant amplitude effect was detectable at an increase as small as +20 mOsm, and maximal amplitude induction was observed with +300 mOsm condition, suggesting higher osmotic challenge may exceed the physiological range (Fig. 2h, Fig. S4). To further validate the responses of molecular circadian clocks to loading and osmolarity, we used cartilage and IVD tissue explants from a different clock reporter mouse model, the Cry1-Luc which is a promoter reporter[27] as opposed to the PER2::Luc fusion protein reporter[28]. The dose-dependent clock resetting effect by both mechanical loading and osmolarity was also observed in cartilage and IVD explants from *Cry1*-Luc mouse (Fig. S5).

## Daily hyperosmotic challenge synchronizes dampened circadian rhythms in both young and aging skeletal tissues

With the onset of the activity phase, mouse cartilage and IVD tissues experience roughly 12 h of loading within a 24-h day, leading to increased osmolarity[22]. Having determined that the acute increase in osmolarity is a clock synchronizing factor, we next used a protocol to approximate the diurnal osmotic cycles experienced by skeletal

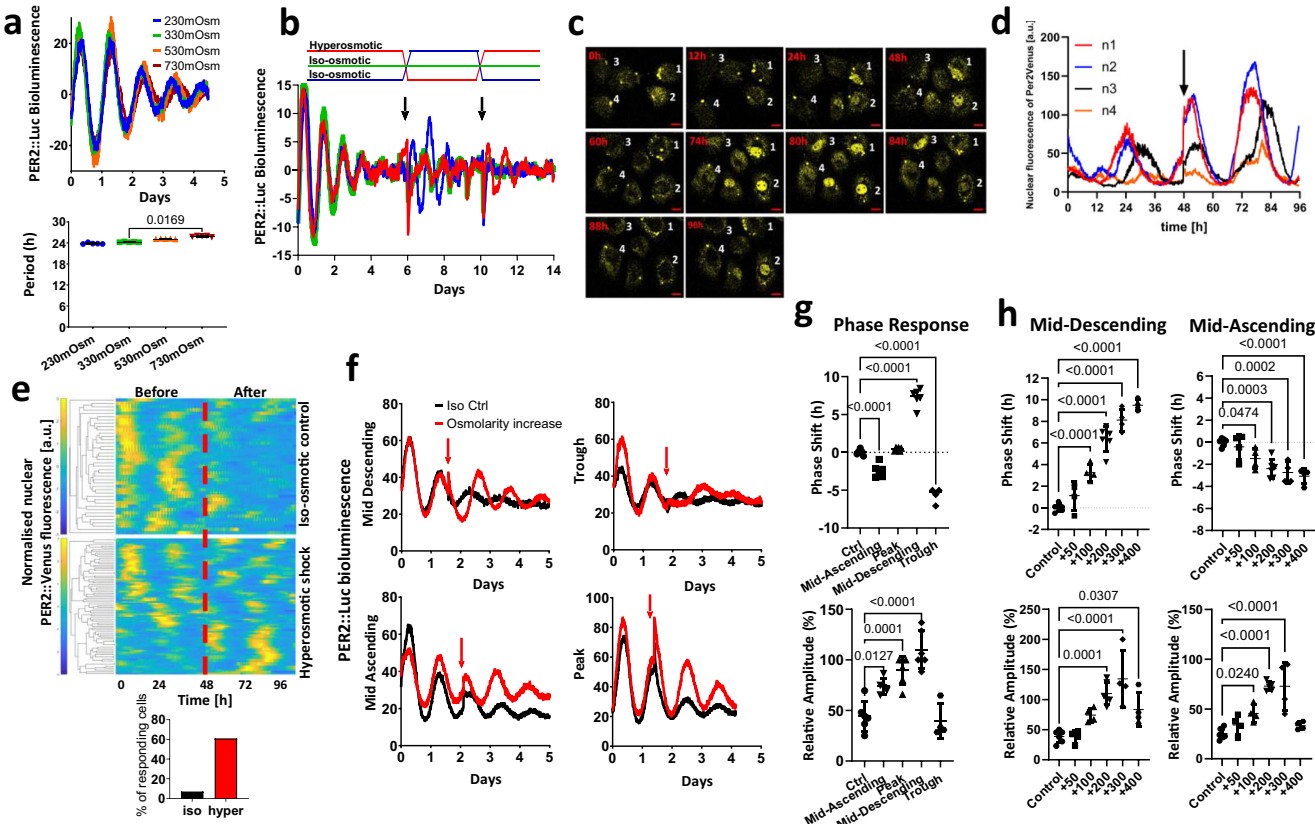

**Fig. 2 | Hyperosmolarity resets the circadian clock in IVDs and cartilage in a phase and dose dependent manner. a** Oscillations and period of PER2::Luc IVD explants cultured in varying baseline osmotic conditions. Mean ± SD, $n = 5$. **b** Oscillations of PER2::Luc cartilage with media changes. Conditioned media were swapped between explants in iso- and hyper- osmotic conditions on day 6, then swapped back on day 10. Control cultures were undisturbed ($n = 3$). **c** Confocal microscopy images of representative mouse articular chondrocytes expressing PER2::Venus (displayed in hours at indicated times). Cells were cultured under normal osmolarity conditions for 48 h, after which osmolarity was increased by +200 mOsm using sorbitol. Scale bar 20 μm. **d** Individual nuclear PER2::Venus trajectories of 4 single cells from (**c**) were plotted in the graph. **e** Heat maps of PER2::Venus single-cell trajectories before and after hyperosmotic treatment normalized to the area under the curve of the first 48 h. The red dotted line indicates time of treatment. Cells were monitored for up to 96 h, $n = 78$ cells per condition. Bottom bar graph shows the percentage of cells in the cell population responding to osmotic treatment. A cell was classified as responding if the amplitude of the peak following treatment was higher than the amplitude of the peak preceding the treatment. **f** Effects of hyperosmolarity (+200 mOsm, applied at 6 h intervals indicated by red arrow) on PER2::Luc IVD oscillations ($n = 6$). **g** Quantification of phase shift and amplitude induction in (**f**). **h** Quantification of dose-dependent phase shift and amplitude induction in IVD explants treated with increasing osmolarity (+50 to +400 mOsm) at mid-descending and mid-ascending phase of PER2::Luc oscillation ($n = 4$). Statistical analysis was performed using one-way ANOVA. *P* values were adjusted for multiple comparisons using Dunnett's multiple comparisons test. Source data are provided as a Source Data file.

tissues by exposing cartilage and IVD explants to 12 h of hyperosmotic condition and returning them to 12 h of iso-osmotic medium. A single cycle with a hyperosmotic challenge as low as +100 mOsm synchronized the circadian rhythm in cartilage (Fig. 3a) and IVDs (Fig. 3b) from young mice (2 months old) that have been in culture for 5 days, although, this effect seems to be dependent on the extent of osmotic change, or the number of cycles applied, or both. When explants were exposed to two daily cycles of +200 mOsm challenges, they showed an even stronger clock amplitude (Fig. 3c, d).

We have previously shown that the circadian amplitude of IVD and cartilage rhythms dampen with aging[18,20]. Therefore, we tested whether clocks in aging skeletal tissues still respond to osmotic cycles. Indeed, exposure of aging explants to two osmotic cycles resulted in resynchronization in both articular cartilage (Fig. 3e) and IVDs (Fig. 3f).

### Hyperosmotic challenge induces rhythmic global gene expression patterns in a cell-type-specific manner

To gain mechanistic insights into the clock resetting mechanisms, we initially investigated how chondrocyte circadian clocks responded to hyperosmotic challenge in vitro. Clock-unsynchronized primary chondrocytes from PER2::Luc mice exhibit only a low amplitude

oscillation. Exposure to +200 mOsm increase in osmolarity augmented PER2::Luc amplitude (Fig. 4a). Similar observations were made in a human IVD annulus fibrosus cell line (Fig. S6a). Strikingly, the same hyper-osmotic stimuli disrupted circadian oscillation in U2OS cells, a human osteosarcoma cell line widely used as a cellular model of circadian clocks, as well as in keratinocytes (Fig. S6b and c). As such, the osmolarity-entrainment of circadian clock is likely cell-type dependent and could indicate cellular adaptation to their local niche.

We next determined to what extent hyperosmolarity can synchronize circadian rhythms of gene expression at the transcriptome level by circadian time-series RNA sequencing in primary mouse chondrocytes. The PER2::Luc reporter allowed us to track clock rhythms in parallel cultures (Fig. 4a). mRNA samples were collected every 4 h for 2 full circadian cycles, starting with samples just before the osmotic increase (ZT 0). Principal component analysis of the RNAseq results revealed the circadian time of sampling as the biggest factor separating the samples (Fig. 4b). Analysis revealed 1312 rhythmic genes using integrated $p < 0.05$ cut-off threshold (254 rhythmic genes with a BHQ < 0.05 cut-off), including most core clock genes (Fig. 4c, d; Supplementary Data 1). Following hyperosmolarity, 1557 genes showed significant differential expression (with >2-fold change) between T0

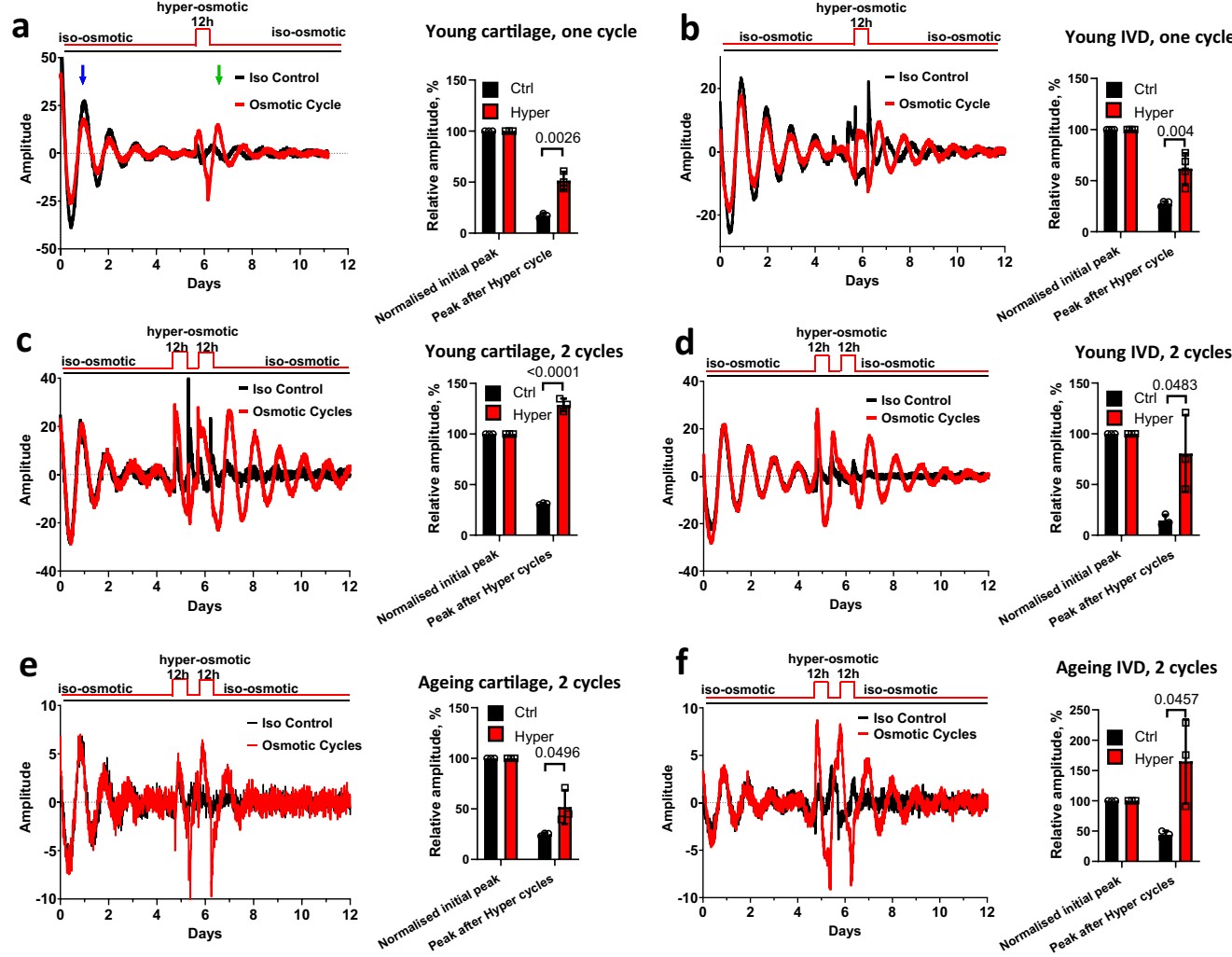

**Fig. 3 | Osmotic cycles synchronize circadian clocks in young and aging cartilage and IVDs. a, b** Effects of one cycle of osmotic changes on PER2::Luc cartilage explants (**a**) and IVDs (**b**) from 2-month old mice. At day 6 explants were exposed to 12 h of +100 mOsm hyperosmotic medium then returned to iso-osmotic media. The amplitude of the peak after osmotic cycle (green arrow) was quantified as % of the peak 24 h after start of recording (blue arrow). **c−f** Effects of two cycles of hyperosmotic challenges (12 h of +200 mOsm/12 h of iso-osmotic media) on cartilage (**c, e**) and IVDs (**d, f**) from 2-month old (**c, d**) or 12-month old (**e, f**) PER2::Luc mice. $n = 3$ in all experiments except 3b $n = 4$. Statistical analysis was performed using two-tailed unpaired $t$ test. Bars represent mean and SD. Source data are provided as a Source Data file.

and T4 (Fig. 4d, Supplementary Data 2). Importantly, most core clock genes appeared as early response genes 4 h after osmotic stress, with significant upregulation of *Bmal1 (Arntl), Tef, Per1, Clock, Cry1/2, Rora* and *Nfil3, and downregulation of Nr1d1, Per2/3, and Dbp* (Fig. 4e).

### Treadmill running elicits transcriptome-wide changes in mouse cartilage and IVD tissues

As previously, mice were gradually adapted to treadmill running for 5 days after which they run at ZT 2 for 45 min for 5 days. Tissues were collected on the last day immediately after treadmill running. Control sedentary littermates were collected at the same time. PCA analysis of the RNAseq results showed clear separation of sedentary and running samples both in cartilage and IVDs (Fig. S7a). Differential expression analysis showed 421 upregulated and 261 downregulated genes in cartilage and 253 upregulated and 470 downregulated in IVDs (Fig. 4f, h, Supplementary Data 7 and 8). Among clock genes *Bmal1 (Arntl)* and *Npas2* were consistently downregulated and *Per1/2, Nr1d2, Tef* and *Dbp* consistently upregulated in both tissues (Fig. 4g, i). Next, we compared the differentially expressed genes in cartilage tissues from treadmill running mice with our published circadian cartilage transcriptome[19]. In

cartilage close to 1/3 of genes differentially expressed in the treadmill experiment were rhythmic in the time-series dataset (Fig. S7b). Most importantly, the expression pattern of these genes in the treadmill running samples which were collected at ZT 2 resembles that of timeseries samples collected at ZT 14-18 (Fig. 4j and Fig. S7c, d), consistent with a global shift of gene expression by exercise timing. Ingenuity Pathway Analysis (IPA) analysis showed Osteoarthritis, HIF1α, circadian rhythm and PI3K/AKT pathways as significantly regulated by both osmotic stress and treadmill running (Fig. 4k and Supplementary Data 3, 5, 9, 11). Upstream regulators common to all three datasets (cartilage run, IVD run, and chondrocyte hyperosmolarity) included BMAL1-Clock complex, p38, ERK, PI3K, AKT, mTOR, TSC2, PP2A as well as CREB, Forskolin, cAMP, Ca²⁺, NFAT5, HSF1, TGFβ and FOXO1, 3 and 4 (Fig. 4l and Supplementary Data 4, 6, 10, 12).

### PLD2-mTORC2-AKT-GSK3β as a convergent pathway for loading- and hyperosmolarity- induced clock resetting

Based on the IPA analysis we selected several candidate pathways for further analysis. Pharmacological inhibition of pathways such as

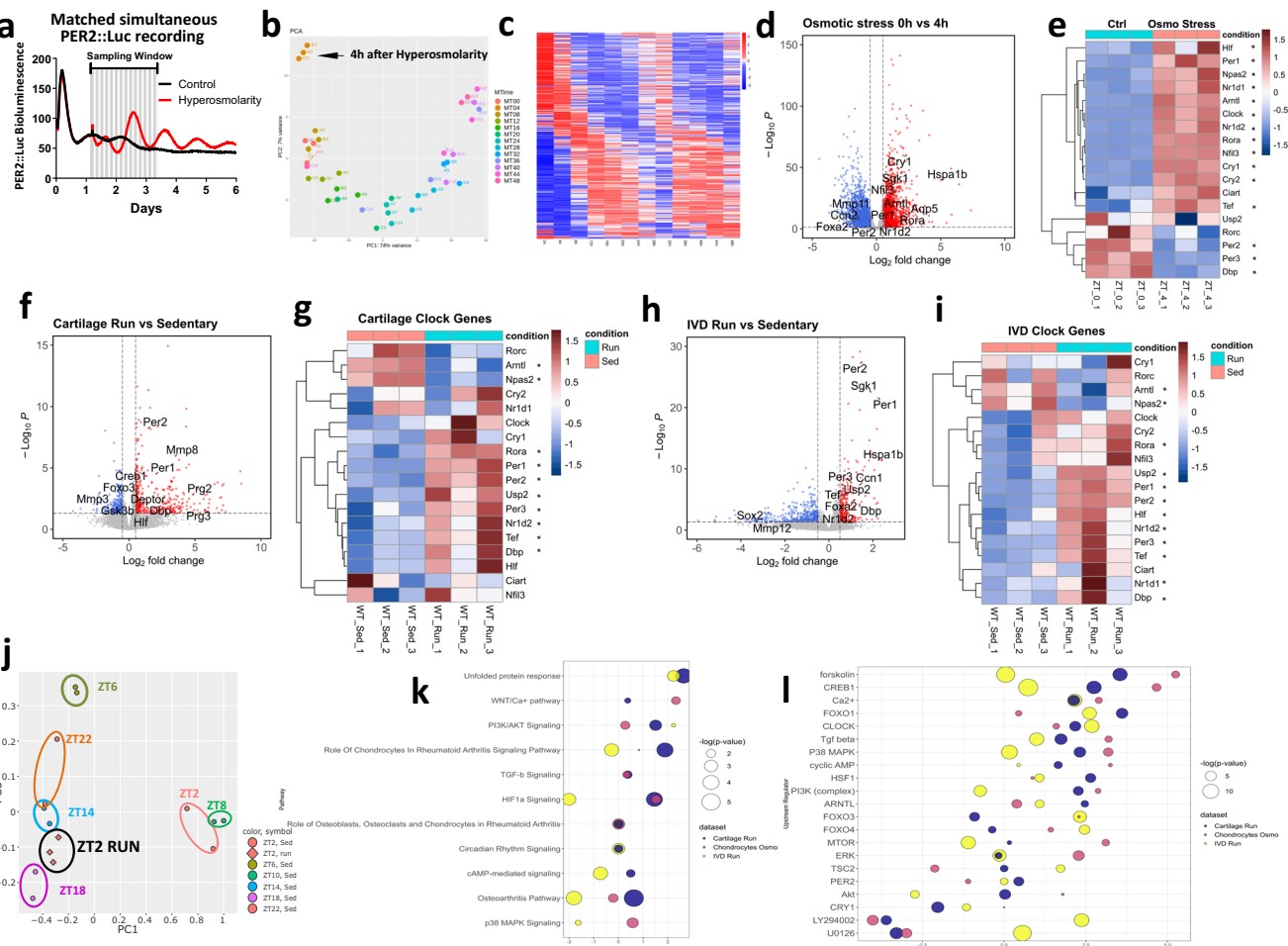

**Fig. 4 | Hyperosmolarity and treadmill running induce transcriptome-wide changes in gene expression. a** Bioluminescence recording of primary chondrocytes isolated from PER2::Luc mice. Cells were not synchronized at the beginning of the experiment. After 30 h, media osmolarity was increased by +200 mOsm. Parallel samples were harvested every 4 h for RNA isolation and RNAseq analysis. **b** Principal component analysis (PCA) showing a correlation between RNAseq replicates and a developing trend over time. **c** Heatmap showing gene expression patterns of 254 rhythmic genes (BHQ < 0.05) following increase of osmolarity. **d** Volcano plot showing differentially expressed genes between time-point 0 (T0) and 4 h (T4) after osmotic stress. 630 were up regulated and 927 downregulated at Adj p < 0.05 and Log₂FC = 1 cut off. **e** Heatmap of circadian clock genes at T0 and T4 after osmotic stress. *p < 0.05. **f–i** Volcano plot showing differentially expressed genes between sedentary and treadmill running mice in cartilage (**f**) and IVD (**h**) at Adj p < 0.05 and Log₂FC = 0.5 cut off. Heatmaps depicts circadian clock genes in cartilage (**g**) and IVDs (**i**) from sedentary and treadmill-running mice. The asterisk denotes significant changes, *p < 0.05. **j** PCA of treadmill-regulated rhythmic genes at ZT2 vs. circadian time series rhythmic genes in cartilage. **k**, **l** Bubble plots showing significant canonical pathways (**k**) and upstream regulators (**l**) by Ingenuity Pathway Analysis in RNAseq datasets from osmotic stress (0 h vs. 4 h) and treadmill exercise (running vs. sedentary cartilage and IVD). P values were adjusted for multiple comparisons from DESeq2 (**d–i**) and IPA analysis (**k**, **l**) were used to generate the plots. Source data are provided in Supplementary Data.

cAMP/CREB, calcium channels, Rho/ROCK, p38 and ERK failed to block clock responses in tissues to osmolarity and mechanical loading (Fig. S8-S11). One pathway that featured prominently in Upstream Regulators analysis was mTOR (Fig. 4l). Therefore, we tested two mTOR inhibitors, Torin1 (blocking both mTORC1 and mTORC2 complex) and Rapamycin (primarily blocking mTORC1). While Torin1 completely blocked the increase in amplitude following hyperosmotic challenge in cartilage and IVD explants, Rapamycin had no effect (Fig. 5a, b), indicating involvement of mTORC2. AKT is a known phosphorylation target of mTORC2[29] and can be activated by hyperosmolarity in renal cells[30]. Indeed, pre-treatment of cartilage explants with an AKT inhibitor prevented the increase in circadian amplitude following hyperosmotic challenge (Fig. 5c). Western blotting showed an increase in phosphorylation of AKT at the mTORC2 site 8 h after an increase in osmolarity. Pre-treatment with Torin1 but not with rapamycin prevented this phosphorylation (Fig. 5d and Fig. S12). AKT is a known upstream regulator of GSK3β activity, a key clock-regulating

kinase implicated in the regulation of the stability and/or nuclear translocation of PER2, CRY2, CLOCK, REV-ERBα and BMAL1[31,32]. Indeed, inhibition of GSK3β activity by lithium had a synchronizing effect on IVD clocks similar to that of hyperosmolarity (Fig. 5e). Concordantly, hyperosmotic challenge increased GSK3β phosphorylation at Ser9 and Ser389 (corresponding to reduced activity) (Fig. 5f). Pre-treatment with Torin1 (but not Rapamycin) or an AKT inhibitor decreased GSK3β phosphorylation at Ser9 and Ser389 following hyperosmotic challenge (Fig. 5f and Fig. S10). We next verified whether the above mechanisms were also involved in clock resetting by mechanical loading. Pre-treatment with the mTORC2 or AKT inhibitor (but not Rapamycin) completely blocked the mechanical loading-induced increase in circadian clock amplitude (Fig. 5g, h, Fig. S10). Consistent with our findings, mechanical strain in mesenchymal stem cells initiates a signaling cascade that is uniquely dependent on mTORC2 activation and phosphorylation of AKT at Ser-473, an effect sufficient to cause inactivation of GSK3β[33].

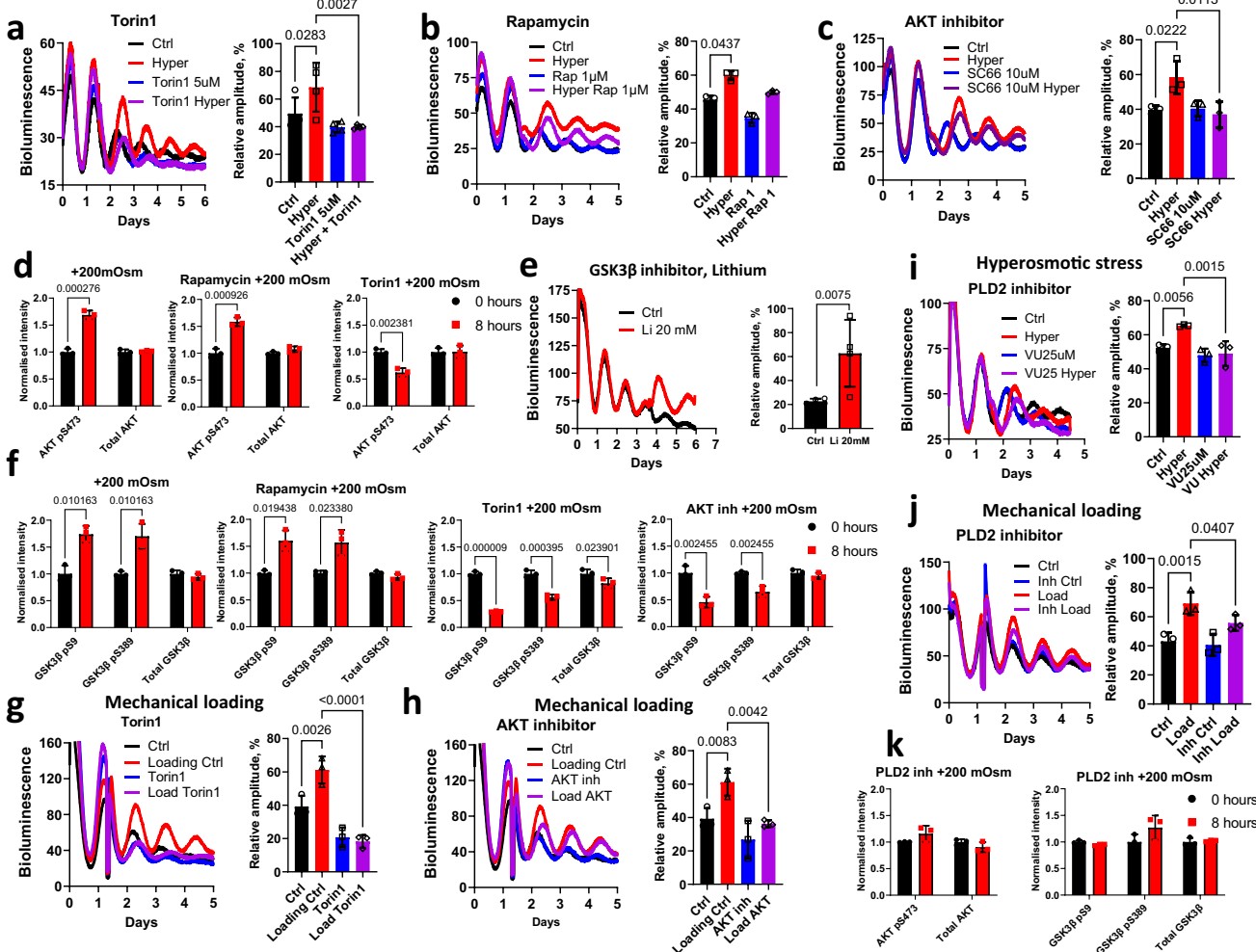

**Fig. 5 | The PLD2-mTORC2-AKT-GSK3β pathway is a convergent mechanism mediating the loading and hyperosmolarity elicited clock entrainment.** **a**–**c** Effects of mTORC1/2 inhibitor Torin1 (**a**), mTORC1 inhibitor Rapamycin (**b**) and AKT inhibitor SC66 (**c**) on blocking the hyperosmolarity (+200 mOsm) -induced clock amplitude change in PER2::Luc cartilage explants. **d** WB quantification showing the effect of Rapamycin or Torin1 on total and phosphorylation levels of AKT at Ser473 in mouse primary chondrocytes 8 h after increased osmolarity. **e** Bioluminescence recording and amplitude quantification of PER2::Luc IVD explants treated with 20 mM lithium (an inhibitor of GSK3β). **f** Effects of Torin1, Rapamycin and the AKT inhibitor SC66 on blocking the hyperosmolarity-induced phosphorylation of GSK3β at Ser9 and Ser379 in mouse primary chondrocytes. **g**, **h** Effects of Torin1 (**g**)

and SC66 (**h**) on blocking the loading (0.5 MPa, 1 Hz, 1 h) induced clock amplitude change in PER2::Luc cartilage explants. **i**, **j** Effect of PLD2 inhibitor on blocking osmolarity (**i**) and mechanical loading (**j**) induced increase in PER2::Luc bioluminescence in mouse cartilage. **k** WB quantification showing the effect of PLD2 inhibitor pre-treatment on total and phosphorylation levels of AKT (at Ser473) and GSK3β (at Ser9 and Ser379) in mouse primary chondrocytes 8 h after increased osmolarity. $n = 3$ in all bioluminescence recording experiments except an $n = 4$. Statistical analysis was performed using one-way ANOVA for (**a**–**c**, **g**–**i**, **j**) (P values were adjusted for multiple comparisons using Dunnett's multiple comparisons test) and Two-tailed unpaired $t$ test for (**d**–**f**, **k**). Source data are provided as a Source Data file.

Finally, the lack of involvement of membrane calcium channels (Fig. S9a-d) and Rho/ROCK pathway (Fig. S9e) prompted us to explore what cell surface mechano-sensors might mediate the clock resetting effects of loading and osmolarity. PLD2 is a plasma membrane tension sensor that connects the mechanical forces involved in cell swelling and shrinking as well as external stimuli with the mTORC-AKT pathway[34,35]. PLD2 catalyses the conversion of PIP2 to Phosphatidic Acid which stabilizes the mTORC complexes[36]. Pre-treatment with the PLD2 inhibitor prevented amplitude increase for both the osmotic stress and mechanical loading of cartilage explants (Fig. 5i, j). Western blotting showed that the PLD2 inhibitor prevented phosphorylation of AKT at the mTORC2 site as well as phosphorylation of GSK3β after hyperosmotic stress (Fig. 5k).

## Discussion

Taken together, we have provided in vivo, ex vivo and in vitro experimental evidence that daily cycles of mechanical loading and

associated rhythmic changes of osmolarity within the physiological range act as bona fide entrainment cues for skeletal circadian clocks.

In mammals, the circadian system is organized as a hierarchy, with the central clock SCN temporally coordinating all peripheral clocks in various organs. The stable phase relationships between the SCN and peripheral tissues render clock timing information useful for the entire multicellular organism[37]. The SCN-controlled sleep-wake cycle sets whole organism-level systemic rhythms. Mammalian body temperature may fluctuate up to 3 °C over a circadian cycle, with increases during active and decreases during the resting phase. This small oscillation can weakly synchronize skeletal clocks where PER2 expression coincides with increase in temperature[18,20]. In contrast to temperature, treadmill running exercise in mice 2 h after the beginning of normal resting phase led to a large phase advance of the skeletal clocks of up to 8 h. Interestingly, this exercise protocol did not change the SCN circadian rhythm, leading to an uncoupling of the central and skeletal clocks. Moreover, direct exposure of ex vivo cartilage and IVD

tissues to mechanical loading or hyperosmolarity was sufficient to induce circadian oscillations, thus excluding the involvement of systemic factors in synchronization of the clock by treadmill running. Based on our findings, we propose these daily physiological inputs as key tissue-specific zeitgebers. This specificity may allow flexibility for cartilage/IVD clocks to be uncoupled from the SCN clock to cope with changing environmental needs. Indeed, when typical nocturnal animals are energetically challenged by food shortage or thermal stress, they switch temporal niche from nocturnal to diurnal[38,39]. In this study, we examined a range of physiologic and hyperphysiologic mechanical loading parameters; however, it is important to note that we are restricted by the current lack of direct measurements of the precise mechanical loading environment in mouse cartilage and IVD in vivo, as well as the limitation of the sensitivity of our loading equipment. Thus, we were not able to test a broader range of loading parameters. Future work is clearly warranted in this regard, including the development and testing of direct mechanical loading systems in vivo to assess the minimal day/night differences required for entrainment of the cartilage and IVD clocks.

We show that the clock resetting effect by loading and hyperosmolarity is circadian phase dependent. An osmotic increase or mechanical loading at the peak of PER2 (which corresponds to the activity onset in mice) does not shift the phase but strengthens the amplitude of skeletal clocks, which could have important implications in exercise timing. A notable aspect of clock network synchrony is its apparent redundancy, with multiple cues signaling timing to specific tissues in a synergistic manner. Regular exercise is thought to be beneficial for the maintenance of skeletal muscle, joint and IVD as well as reducing severity of osteoarthritis symptoms[17,40]. However, no critical studies have directly evaluated the importance of time-of-day. Importantly, IPA of differentially expressed genes in our hyperosmolarity and treadmill running RNAseq datasets showed osteoarthritis pathway as one of the most significantly affected pathways in cartilage, hinting at potential detrimental effects of exercising at an inappropriate time. The magnitude of mechanical loading may also be of importance as we have shown disruption of circadian rhythms by abnormal loading in skeletal tissues[41]. Supporting the importance of intervention timing, mis-timed food intake can have severe adverse metabolic effects in animal studies and in human shift workers, while caloric restriction, when aligned with the circadian timing, brings additional benefits in further extending the lifespan of mice by a striking 35%[42–44]. Cartilage and IVD circadian rhythms are known to be sensitive to age-related dampening and genetic ablation of the clock results in progressive arthropathy[22]. However, we found that repeated daily osmotic cycles lead to resynchronization of circadian rhythm in aging tissue explants. Therefore, interventions reinforcing rest/activity patterns in aging individuals could potentially contribute to skeletal health through strengthening the amplitude of circadian oscillations in gene expression. Our findings also place PLD2, mTORC2, AKT, and GSK3β as key players in skeletal tissue responses to exercise. These molecules are known to play key roles in autophagy, protein synthesis, cell survival and glycogen synthesis, suggesting these processes may also exhibit a circadian pattern. Our data implicate PLD2-mTORC2-AKT-GSK3β pathway as a critical converging mechanism for clock responses to both mechanical loading and osmolarity. Supporting our findings of underlying pathways, mTOR-AKT signaling has recently been implicated in cellular clock phase shifting triggered by hyperosmotic stress in cultured fibroblasts[45]. However, subtle differences exist in downstream signaling pathways in response to these two stimuli. Namely, inhibitors of p38 and ERK1/2 were able to block hyperosmolarity-induced synchronization but had no effect on the response to mechanical loading. Notably, a study by Imamura et al found ASK family kinases to play a crucial role in resetting of the fibroblast circadian clock by changes in osmolarity[46]. However, in their in vitro studies cells exhibited type 0 resetting and reacted to both

hyper and hypo osmotic conditions while our tissue explants showed type 1 resetting at +200 mOsm challenge and reacted to hyperosmolarity only, suggesting cell type and tissue-specific clock responses. These biological insights may help us understand the impact aging and sedentary lifestyle have on daily changes in the physiology of skeletal tissues and facilitate the design of time-prescribed interventions to maintain musculoskeletal health.

## Methods

### Animals
All animal studies were performed in accordance with the 1986 UK Home Office Animal Procedures Act. Approval was provided by the University of Manchester's Animal Welfare and Ethical Review Board (AWERB). Mice were housed in Tecniplast Green Line cages and racks at 20–22 °C and average 60% humidity, with standard rodent chow available *ad libitum* and under 12:12 h light-dark schedule (light on at 7 am; light off at 7 pm). To investigate tissue-level responses to treadmill exercise in vivo and mechanical loading/osmotic challenge ex vivo, PER2::Luc, PER2-Venus and *Cry1*-Luc reporter mouse lines were chosen because (1) these well-established clock gene reporters represent the negative arm of the molecular clock which has been shown to be highly rhythmic and is a faithful reflection of the internal tissue clocks; and (2) they represent different levels of clock gene regulation, with PER2::Luc and PER2-Venus being protein fusions (protein level reporter), while *Cry1*-Luc being promoter driven (transcriptional reporter). The PER2::Luc mice carry the firefly luciferase gene fused in-frame with the 3′ end of the *Per2* gene, creating a fusion protein reporter[28]. *Cry1*-Luc mice carry the luciferase transgene under *Cry1* promoter control and were previously described[27]. PER2::Venus mice carry the Venus fluorescent protein fused with the endogenous PER2 protein and were previously described[47]. For the majority of experiments requiring cartilage and IVD tissue collection, mice were sacrificed by raising $CO_2$ concentration. In experiments requiring SCN collection, mice were sacrificed by cervical dislocation.

### Treadmill exercise
Mice were exercised on TSE Treadmill System setup with eight individual running lanes. Treadmill running was performed 2 h after lights on according to the following protocol:

Day 1: Mice were placed in the treadmill without the motor turned on for acclimatization for about 5 min, following which the power was turned on and the mice allowed to acclimate to the mechanical noise for another 5 min. Next, mice walked at a 10-degree incline at 5 m/min for 10 min. Control group was moved to the same room but remained in their cages.

Day 2: 2 min warm up at start 8 m/min, then speed was increased to 10 m/min for 15 min.

Day 3: 2 min warm up at start 8 m/min followed by 10 m/min gradually increasing to 15 m/min over 20 min.

Day 4: 2 min warm up at start 8 m/min, then 15 m/min for 30 min.

Day 5: 2 min warm up at start 8 m/min, then 15 m/min for 40

Day 6,7: Rest and recovery.

Day 8–12: 2 min warm up at start 8 m/min, then 15 m/min for 45 minutes.

Mice were sacrificed by Shedule1 cervical dislocation after treadmill running on day 12 and tissues were harvested for explant culture or RNAseq.

### Tissue explant culture, mechanical loading, and osmotic challenge
Mouse articular cartilage tissue explants were prepared as described before[19] by dissecting the whole hip joint and then detaching the cartilage from subchondral bone of the femoral head using a scalpel. From the same mice whole IVDs were dissected from the lumbar region of the spine as we described before[20]. Briefly, the whole spines

were dissected and cleaned of muscle tissue using scissors. A scalpel was gently wedged between the vertebral body and the IVD until the tissues separated. Explants were cultured and recorded at 37 °C, 5% $CO_2$ in DMEM/F12 without phenol red and without serum. The osmolarity of the medium was increased using sorbitol and measured using freezing point osmometer. In baseline osmolarity and the conditioned media swap experiments (Fig. 2a, b and S2b), tissue explants were adapted to the indicated media osmolarity for 3 days, after which dexamethasone (100 nM) was added for 1 h to re-synchronize the tissues to the same phase. Next, culture media were changed to recording medium (culturing medium + 100 μM luciferin) of the same osmolarity as adapted to. Tissue explants were recorded either undisturbed (Fig. 2a) or the conditioned media was swapped between dishes (Fig. 2b and S2b). In all other osmotic treatment experiments, 2 M sorbitol dissolved in a culture medium was added to recording dishes at a volume required to achieve indicated osmolarity. An equal volume of culture medium without sorbitol was added to control dishes. Mechanical loading of tissue explants was performed in Bio-Press compression plates on FX-5000 system (FlexCell). Explant tissues were placed in the recording medium and their initial phase recorded in the Lumicycle. At the indicated time all explants were removed from recording dishes and placed in the BioPress plates in a fresh culture medium. Explants were subjected to loading protocol while control explants were placed in the same incubator as the compression instrument but not subjected to loading. SCN brain slices from PER::LUC mice were prepared as described previously[48]. Briefly, mice were culled by cervical dislocation, brains were then removed and moistened with ice-cold HBSS (Sigma) supplemented with 0.035% sodium bicarbonate (Sigma), 10 mm HEPES (Sigma), and 100 μg/ml penicillin-streptomycin (Gibco Invitrogen). Coronal brain slices (250-μm thick) were cut using a vibroslicer (Camden Instruments) and transferred to sterile tissue culture dishes (Corning) in cold HBSS. Using a dissecting microscope, the SCN were identified and microdissected before excised unilateral SCN nuclei were cultured on interface-style cell culture inserts (PICM0RG50; Millipore) in standard 35-mm plastic-based cultures dishes (Corning). One milliliter of sterile culture medium [DMEM (D-2902; Sigma) supplemented with 3.5 g/L d-glucose (Sigma), 0.035% sodium bicarbonate (Sigma), 10 mm HEPES buffer (Sigma), 25 μg/ml penicillin-streptomycin (Gibco), B27 (Invitrogen) and 0.1 mm luciferin (Promega) in autoclaved Milli-Q water] was used per culture.

## Bioluminescence recording and imaging

Bioluminescence from PER2::Luc tissue explants or clock reporter cells were recorded in real-time in Lumicycle (Actimetrics) as described before[19]. Briefly, the cells or tissues were placed in 10 mm petri dishes covered with glass cover slips and sealed with vacuum grease. For recording, 100 μM luciferin was added to the medium. When required, explants were synchronized using 100 nM dexamethasone for 1 h before bioluminescence recording. Where appropriate traces were normalized using 24 h moving average. Live tissue bioluminescence of PER2::Luc IVD explant was imaged using a self-contained Olympus LuminoView LV200 microscope and recorded using a cooled Hamamatsu ImageEM C9100-13 EM-CCD camera[19]. Images were taken every hour for 6 days and combined using ImageJ software (NIH).

## Mouse primary chondrocyte culture and cell lines

Primary chondrocytes from 5-day-old mice were isolated according to a published protocol[49]. Briefly, 5-day-old C57bl/6 PER2::Luc mice were sacrificed by decapitation. Knee, hip and shoulder joints were dissected, and any soft tissue removed. Joint cartilage was subjected to pre-digestion with collagenase D (3 mg/mL) in DMEM two times for 30 min at 37 °C with an intermittent vortex to remove soft tissue leftovers. Subsequently, cartilage was diced using a scalpel and digested overnight at 37 °C. Cells were dispersed by pipetting and passed

through 70 μM cell strainer. Cell suspension was then centrifuged and the pellet was re-suspended in DMEM/F12 with 10% FBS and plated in T75 flasks. Cells were passaged only once before experiments. Immortalized human nucleus pulposus (NP) cell line NP115 was kindly gifted by GC Van Den Akker, TJ Welting, and JW Voncken (Maastricht University, Netherlands) and described previously[50]. The cell line was stably transfected with a pT2A-Per2-Luc reporter construct (a kind gift from Kazuhiro Yagita, Kyoto Prefectural University of Medicine) using nucleofection, followed by single-cell colony selection. U2OS cells stably transfected with Per2-Luc plasmids were kindly gifted by Patrick Nolan (MRC Harwell). HaCaT cells were kindly gifted by Talveen Purba (University of Manchester).

## Calcium imaging of primary chondrocytes

2 days before calcium imaging, primary chondrocytes were plated onto 4 chamber 35-mm glass-bottomed dishes (Greiner Bio-One). 30 min before imaging medium was changed to 500 μL DMEM/F12 without FBS (with or without 1 mM calcium). Next, 0.5 μL of 1 mM Fluo-4 AM calcium dye (Thermo Fisher) was added to each chamber of cells before incubation on the Zeiss Exciter confocal microscope stage at 37 °C in humidified 5% CO2. Imaging was performed using 488 nm excitation wavelength and 520 nm band-pass filter for emission and Fluar 40× NA 1.3 (oil immersion) objective. Image capture was performed with the Zeiss software Aim version 4.2 utilizing the Autofocus macro[51]. Fluorescence analysis was performed on all cells in the field of view using the Zeiss software Aim version 4.2.

## PER2-Venus live cell imaging

Mouse articular chondrocytes were plated onto 35 mm-glass-bottomed dishes (Greiner Bio-One) and sealed with a 40-mm glass coverslip (Thermo Fisher Scientific) and high vacuum grease (Dow Corning) 2 days prior to the experiment and incubated on the microscope stage at 37 °C in humidified 5% $CO_2$. Carl Zeiss LSM880 AxioObserver confocal microscope was used with Fluar ×40/1.30 M27 Oil objective. The 514 nm (ATOF set at 6%) line from an argon ion laser was used to excite the PER2::Venus fusion protein and emitted light between 520 and 550 nm was detected through pinholes set to 5 μm. For the series of interrelated confocal experiments, the same microscope settings have been used. Image capture was performed using the Zeiss Zen2 software. Images were taken every 10 minutes for 96 h. After 48 h of image capturing, the osmolarity of the culture medium was increased by +200 mOsm using sorbitol and imaging continued for another 48 h. Quantification of PER2::Venus nuclear fluorescence was performed using tracking of cell nuclei with Cell Tracker (version 0.6)[52]. The data was exported as mean fluorescence intensity. Trajectories of the nuclear PER2::Venus were normalized across presented conditions and displayed as heat maps. Heat maps were produced using a clustergram function in Matlab R2020a transforming trajectories across cells to 0 mean and standard deviation 1.

## Cyclic mechanical loading on rat NP cell line

Immortalized rat NP cells were kindly generated and donated by Di Chen (Shenzhen Institutes of Advanced Technology) and maintained in DMEM/F-12 (1:1) (Gibco, USA) containing 10% FBS (Invitrogen, USA) and 1% penicillin/streptomycin (Gibco) at 37 °C under 5% $CO^2$ and 20% $O^2$[53]. Compression was conducted 24 h after cell attachment. The rat NP cells were divided into two groups and held under antiphase compression cycles in compression culture chamber (Taikang Biological Technology, Xi'an, China) (alternating 12:12 h cycles of 0.5 MPa/ 0 MPa for 3 days). After compression cycles, the two groups of NP cells were cultured in conventional incubator and mRNA was extracted every 4 h for 48 h before RNA isolation and qRT-PCR. Primers sequence (5′−3′, Rattus norvegicus):

Bmal1 forward, GACTTCGCCTCCACCTGTTCAA;
Bmal1 reverse, GCAGCCCTCATTGTCTGGTTCA.

*Cry1* forward, GCGGAAACTGCTCTCAAGGA;
*Cry1* reverse, CCCGCATGCTTTCGTATCAG.

## RNAseq of mouse tissues and primary chondrocytes

Mouse primary chondrocytes were passaged into 6-well plates. Upon reaching confluency, media were changed to fresh and ~30 h later osmolarity was increased by +200 mOsm using sorbitol. T0 sample was collected just before the addition of sorbitol. Then samples were collected every 4 h for 48 h. mRNA was extracted using RNeasy micro kit (Qiagen) according to the manufacturer's protocol. Quality and integrity of the RNA samples were assessed using a 2200 TapeStation (Agilent Technologies). Libraries were generated by the Genomic Technologies Core Facility using the TruSeq® Stranded mRNA assay (Illumina, Inc.) according to the manufacturer's protocol. Briefly, total RNA (0.1-4 µg) was used as input material from which polyadenylated mRNA was purified using poly-T, oligo-attached, magnetic beads. The mRNA was then fragmented using divalent cations under elevated temperature and then reverse transcribed into first-strand cDNA using random primers. The second strand cDNA was then synthesized using DNA Polymerase I and RNase H. Following a single 'A' base addition, adapters were ligated to the cDNA fragments, and the products then purified and enriched by PCR to create the final cDNA library. Adapter indices were used to multiplex libraries, which were pooled prior to cluster generation using a cBot instrument. The loaded flow-cell was then paired-end sequenced (76 + 76 cycles, plus indices) on an Illumina HiSeq4000 instrument. Finally, the output data was demultiplexed (allowing one mismatch) and BCL-to-Fastq conversion performed using Illumina's bcl2fastq software, version 2.20.0.422.

## Bio-informatic analysis

Unmapped paired-end sequences were tested by FastQC (http://www.bioinformatics.babraham.ac.uk/projects/fastqc/). Sequence adapters were removed and reads were quality trimmed using Trimmomatic_0.39[54]. The reads were mapped against the reference mouse genome (mm10/GRCm38) and counts per gene were calculated using annotation from GENCODE M21 (http://www.gencodegenes.org/) using STAR_2.7.7a[55]. Normalization and differential expression were calculated with DESeq2_1.28.1[56]. Only genes exceeding 50 counts in at least one timepoint were used subsequently. Principal Components Analysis was done using sklearn algorithm (scikit-learn Python library). Analysis of rhythmic genes was done with MetaCycle_1.2.0[57]. MetaCycle is an algorithm for the detection of rhythmicity that combines results of three methods (Lomb-Scargle periodograms, JTK_CYCLE and Arser) that involve least-squares fits to sinusoidal curves, detecting monotonic orderings of data across ordered independent groups and autoregressive spectral estimation. In subsequent analysis, we used the Meta_2d function of MetaCycle which provides integrated p value of the three methods as well as BHQ adjusted p value. For Ingenuity Pathway Analysis (Qiagen) we used the integrated p value list to investigate rhythmic pathways. Differentially expressed genes between T0 and T4 were used for Upstream Regulator analysis with threshold set at adjusted $p < 0.05$ and $-Log_2$ Fold Change $\leq 1$ and $\leq 0.5$ for treadmill running tissues.

## Western blotting

Western blotting was performed according to standard procedures. All primary antibodies were purchased from Cell Signaling Technology ERK1/2 (#4695), pThr202/pTyr204 ERK1/2 (#4370), p38 (#8690), pThr180/pTyr182 p38 (#4511), GSK3β (#12456), pSer9 GSK3β (#5558), except for pSer389 GSK3β (14850-1-AP) which was purchased from Proteintech and αTubulin (T9026) purchased from Sigma Aldrich. Primary antibodies were used in 1:1000 dilution. Secondary antibodies: LI-COR IRDye® 800CW Goat anti-Mouse IgG (926-32210) and IRDye® 680RD Goat anti-Rabbit IgG (926-68071) were used in 1:20,000

dilution. WB was imaged using the LI-COR Odyssey Imaging System and quantified using Image Studio Lite 5.2.

## Reagents

Dexamethasone and Forskolin were purchased from Sigma. Inhibitors Y-27632, 666-15, PD184352, SB203580, Rapamycin, Torin1, SC66 and VU 0364739 were purchased from Tocris. KG-501 and GDC-0994 were purchased from Cambridge Bioscience Ltd.

## Statistical analysis

Data were evaluated using two-tailed Student's *t* test or One way ANOVA. Following ANOVA individual comparisons were performed and *p* values were adjusted for multiple comparisons using Dunnett's multiple comparisons test. Results were presented as mean ± SD from at least three independent experiments. Differences were considered significant at the values of $P < 0.05$.

## Reporting summary

Further information on research design is available in the Nature Portfolio Reporting Summary linked to this article.

## Data availability

RNAseq raw data were deposited to ArrayExpress repository (accession no. E-MTAB-11040 for osmotic stress in primary chondrocytes and E-MTAB-12877 for treadmill running tissues). Circadian cartilage RNAseq dataset E-MTAB-3428 was used for comparison with the treadmill running dataset. All other data are contained within the manuscript and supplementary information. Source data are provided with this paper.

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

## Acknowledgements

The authors thank J Takahashi (UT Southwestern Medical Center, US) and M Hastings (MRC LMB, Cambridge, UK) for the PER2::Luc and *Cry1*-Luc mouse lines. We thank the Genomics Core Facility (A Hayes and L Zeef) and the Bioimaging Facility (D Spiller) at the University of Manchester for their kind assistance with RNAseq and imaging studies. We thank R Lucas and J Allen for their comments on our manuscript. This work was funded by MRC project grants MR/T016744/1 and MR/P010709/1 (Q.J.M., J.A.H.); a Versus Arthritis Senior Fellowship Award 20875 (Q.J.M.); a Wellcome Trust Grant for the Wellcome Centre for Cell-Matrix Research 088785/Z/09 (Q.J.M.); an FBMH Facilitating Excellence Fund (M.D.); BBSRC sLoLa grant (BB/T001984/1 to Q.J.M.); National Natural Science Foundation of China grants 82020108019 and 82130070 (Z.J.L. and L.Y.); National Institutes of Health grants AG46927, AG15768, AR080902, AR072999, AR073752, and AR074992 (F.G.).

## Author contributions

Conceptualization: Q.J.M., J.A.H., M.D., Methodology: M.D., D.P., C.L., D.W., L.Y., F.G., Investigation: M.D., D.P., B.B.-O., A.P., N.R., C.G., D.W., C.L., L.Y., Z.J.L., F.G., Q.J.M., J.A.H., Visualization: M.D., A.P., C.G., C.L., D.W. Funding acquisition: QJ.M., J.A.H. Project administration: Q.J.M., J.A.H., Supervision: Q.J.M., J.A.H., Writing—original draft: Q.J.M., M.D., J.A.H., Writing—review & editing: D.P., L.Y., Z.J.L., D.W., F.G., C.G.

## Competing interests

The authors declare no competing interests.
