## [Peer Review File · Nature Communications]

Mechanical loading and hyperosmolarity as a daily resetting cue for skeletal circadian clocksREVIEWER COMMENTS

Reviewer #2 (Remarks to the Author):

In this manuscript, the authors found that the daily cycles of mechanical loading and the resulting cyclic hyperosmotic pressure in rodent cartilage and intervertebral disc (IVD) cause rhythm amplification of the skeletal circadian clock. By using cultured chondrocytes prepared from PER2-Luc mice, they found that a signaling pathway of PLD2-mTORC2-AKT-GSK3beta plays a central role for the hyperosmolarity-induced amplification of the circadian clock. As a key entrainment factor, light/dark cycle synchronizes the central clock in the hypothalamic SCN, while the feeding regulates oscillation of peripheral clocks in peripheral tissues such as liver, and the body temperature fluctuation is also known to regulate the peripheral clocks even in homeotherm. As another synchronizer, the authors report that the mechanical loading during the half of the day is a new tissue niche-specific entrainment factor in skeletal tissues. This study provides an idea that each peripheral tissue can be controlled by not only the SCN but also specific environmental signals. I think this study significantly contributes to our understanding of how the circadian alignment of our daily activities regulates the circadian clock system normally in the body, but several important issues need to be addressed.

1) In treadmill experiments on mice (Fig.1B, C), the forced exercise during the rest phase remarkably shifted the Per2-luc rhythms in the mouse cartilage and IVD but not the SCN rhythm. However, as the authors stated in the text, the force exercise includes metabolic and systemic effects in addition to the loading effects on the skeletal tissues. Accordingly, the loading experiment on the isolated tissues in vitro is absolutely required (Fig.1D-F) to conclude the importance of the mechanical loading for the clock regulation. In humans, the cartilage and IVD experience 2-3.5 MPa during the active phase, followed by low-load of 0.1 MPa during the rest phase (as stated in p.3, line 22). In the mouse tissues experiment (Fig.1D-E), the cartilage and IVD explants were subjected to 1Hz, 0.5MPa compression for one hour. The intensity of 0.5MPa is within a physiological range for human tissues, but it appears too strong for mice. The compression (dose)-dependency on the rhythm (Fig.1G) suggests that the effect on the rhythm may become negligible in a weaker physiological range in mice (e.g., less than 0.05MPa). It is necessary to demonstrate that the skeletal

rhythms can be regulated by the daily changes in compression at physiological intensities in the mouse skeletal tissue.

2) RNA rhythm data of *Bmal1* and *Cry1* genes (Fig.1J) are fluctuating and not clear to demonstrate antiphase gene expression rhythms between IVD cells that were subjected to compression cycles with antiphase. Why did not the authors examine more strongly rhythmic genes such as *Per2* (as shown in Fig.1A)?

3) The experimental procedures for measurement of cellular rhythms are not clearly stated; "PER2::Luc IVD explants were cultured under static osmotic conditions of 230-730 mOsm for 3 days before clock synchronisation and recording" (p.7, line 16-). Did they synchronize cells by DEX "pulse" treatment? Which culture medium was used after synchronization? Did they use a medium with normal osmolarity after synchronization? If so, these data do not demonstrate that "the circadian period in these skeletal tissues is osmolarity-compensated (Fig.2A)". Most probably, the osmolarity has an obvious effect on the cellular rhythms as demonstrated in Fig.2B, and the rhythms are "not compensated". Rather, the Fig.1A data indicate that (potential) changes in circadian period in hyperosmotic medium is "reversible" when returned back to the normal medium.

Similarly, procedures for Fig.2B is unclear: "...and allowed to adapt for 3 days before synchronisation and recording." (p.8, line 1) Did they synchronize cells by DEX "pulse" ? In Fig.2B, the waveform in control in black is invisible, and hence the blue and red lines cannot be compared with the control.

4) It is not clear to me how the results of single-cell imaging in Fig.2C, D, E are related to the other part of the manuscript. In addition, I cannot find detailed explanation for preparation of the heatmap in Fig.1E and how the bottom bar graph was prepared.

5) Phase transition curve (Fig.S2D) should be plotted in CT for both the pulse time and new phase. Otherwise, the slope is not 1 for type 1 resetting.

6) In Fig. 2, the phase-shift is very large (almost up to 10 hours) in +400 experiment. Does this stimulus still show a type-1 resetting? or, does the type 1 phase-shift change to type-0

resetting in response to stronger stimuli?

7) Why do the authors demonstrate the results with a different reporter Cry1-luc mice in Fig.S3? Apparently, the data are slightly different from those obtained in Per2-luc mice, but no explanation is given for the experiments in Fig.S3.

8) In experiments in Fig.3, they compared the effect of one daily cycle of +100 mOsm treatment with two daily cycles of +200 mOsm. Therefore, it is not concluded whether the number of cycles was effective or the strength of the osmolarity was effective (or both). It is required to revise the following statement; .. this effect was clearly dependent on the extent of osmotic change and number of cycles applied. (p.10, line 16)

9) The authors concluded that cellular rhythms from U2OS in culture was not sensitive to application of osmolarity change, but the experiment was performed only at a single time point of the day (Fig.S4B). Because the osmolarity-sensitivity was not observed at a certain time of the day (at Per2 trough) even in skeletal tissues (Fig.2F), it is not easy to conclude that “the osmolarity-entrainment is clearly cell type dependent” from the current experimental data (as stated in p.12, line 12).

10) In the principal component analysis of the RNA seq data, the three replicates of control Sedentary group are broadly scattered as compared with those of the treadmill Running group in both the cartilage and IVD samples (Fig.S5A). What is the possible reason for this.

11) The pharmacological experiments in Fig.5 revealed that PLD2-AKT-mTORC2 is involved in osmolarity-sensitive clock regulation. Interestingly, phosphorylation level of GSK3beta is affected by the osmolarity change, but inhibition of GSK3beta by lithium treatment did not weaken the osmolarity-sensitive amplification, but it rather strengthened the rhythms (Fig.5E). Therefore, change in GSK3beta phosphorylation upon osmolarity change may be irrelevant to the clock regulation. Please clarify the phrase; “as well as phosphorylation of GSK3beta after hyperosmotic stress (Fig 5K)”. (p. 12. line 9)

12) Literature search revealed a recent paper reporting that mTORC2-AKT is responsible for

resetting of fibroblast clock triggered by osmotic stress (doi: 10.1089/ars.2021.0059.)

Together, it appears that the osmolarity-sensitive regulation of cellular clock is common to a variety of cells, and this idea is diverged from the authors' speculation that the signaling pathway is cell-type-dependent (Comment 9). Information about the expression levels of PLD2, AKT or mTORC2 among tissues/cells should help discussion about the issue.

Minor points;

IPA should be spelled out in p.13, line 17. Then, Ingenuity Pathway Analysis (p.19, line 10) can be abbreviated.

Reviewer #3 (Remarks to the Author):

These data support the central premise that mechanical force might be an entrainment factor for cartilage, as compelling data indicate augmentation of different elements of rhythmicity and clock genes in response to force and changes in osmolarity.

The use of both in vivo and ex vivo (uncoupling systemic signals) is a real strength here.

Minor: It might be helpful for the non-expert reader to be reminded why the Per2 reporter is such a relevant read-out here.

Equally, the data strongly support that acute hypertonic shock augments rhythmicity in a manner similar to effects of loading.

Minor: It might again be useful for non-expert to point out that cartilage has a high fixed charge density supporting such fluctuations with loading.

It is slightly hard for the reader to make direct comparisons between responses to loading and hyperosmotic shock (and thus access relative "phenocopy") but nevertheless the effects do appear similar in nature. Perhaps some side-by side analysis in supplementaries would help. The authors do clearly state differences (for example in amplitude effects and later in molecular mechanisms supporting – e.g ERK signalling).

The similarities and physiologically relevant association of force and osmolarity in cartilage rationalise dissecting the effects using agnostic sequencing to find a point of convergence. The identification and detailed dissection of the shared molecular mechanism is a final strength of the study. This reviewer congratulates the researchers on successfully finding this after chasing multiple avenues. A central challenge here was always going to be supporting the notion that the effects of mechanical force on cycling are mediated by a tonic flux given inhibiting the later during the application of load is technically challenging. The hunt for a shared 'sensor' i guess continues.

This is certainly not suggested as a necessary experiment, but do the authors think, or perhaps have any data that speaks to the idea, that old or diseased cartilage, with degraded negatively charged proteoglycan, might exhibit a differential response to loading due to an altered (diminished?) capacity for an osmotic flux? Perhaps the results, as for osmolarity shown in the manuscript, will be the same- as the changes are at the level of cells machinery to sense/transduce environmental changes. It will of course be hard/impossible to uncouple the confounding effects of other changes in the (old/diseased) tissue. Similarly, perhaps an approach using “engineered” tissue/scaffolds with different charge properties, seeded with PER2 reporting chondrocytes might illuminate strengthen further the link between force and osmotic changes.

This work significantly contextualises much of the earlier work by this group in this arena and the thoughtful discussion frames some of the future avenues that might be explored to translate these cell and tissue phenomena to tissue health context. As written, the work certainly supports the conclusions drawn and I see no flaws in design or interpretation prohibiting publication. The group have previously shown the strengths of the methodology used, which is highly appropriate and is all the stronger for the multiple tissue types and primary cell types used. Description of methods relies heavily on their former description when used previously but as written the reader can follow what is being done.

REVIEWER COMMENTS

Reviewer #2 (Remarks to the Author):

In this manuscript, the authors found that the daily cycles of mechanical loading and the resulting cyclic hyperosmotic pressure in rodent cartilage and intervertebral disc (IVD) cause rhythm amplification of the skeletal circadian clock. By using cultured chondrocytes prepared from PER2-Luc mice, they found that a signaling pathway of PLD2-mTORC2-AKT-GSK3beta plays a central role for the hyperosmolarity-induced amplification of the circadian clock. As a key entrainment factor, light/dark cycle synchronizes the central clock in the hypothalamic SCN, while the feeding regulates oscillation of peripheral clocks in peripheral tissues such as liver, and the body temperature fluctuation is also known to regulate the peripheral clocks even in homeotherm. As another synchronizer, the authors report that the mechanical loading during the half of the day is a new tissue niche-specific entrainment factor in skeletal tissues. This study provides an idea that each peripheral tissue can be controlled by not only the SCN but also specific environmental signals. I think this study significantly contributes to our understanding of how the circadian alignment of our daily activities regulates the circadian clock system normally in the body, but several important issues need to be addressed.

We really appreciate this reviewer's kind support of our work and their very thoughtful and constructive comments to improve our manuscript. We are very happy to address the issues below.

1) In treadmill experiments on mice (Fig.1B, C), the forced exercise during the rest phase remarkably shifted the Per2-luc rhythms in the mouse cartilage and IVD but not the SCN rhythm. However, as the authors stated in the text, the force exercise includes metabolic and systemic effects in addition to the loading effects on the skeletal tissues. Accordingly, the loading experiment on the isolated tissues in vitro is absolutely required (Fig.1D-F) to conclude the importance of the mechanical loading for the clock regulation. In humans, the cartilage and IVD experience 2-3.5 MPa during the active phase, followed by low-load of 0.1 MPa during the rest phase (as stated in p.3, line 22). In the mouse tissues experiment (Fig.1D-E), the cartilage and IVD explants were subjected to 1Hz, 0.5MPa compression for one hour. The intensity of 0.5MPa is within a physiological range for human tissues, but it appears too strong for mice. The compression (dose)-dependency on the rhythm (Fig.1G) suggests that the effect on the rhythm may become negligible in a weaker physiological range in mice (e.g., less than 0.05MPa). It is necessary to demonstrate that the skeletal rhythms can be regulated by the daily changes in compression at physiological intensities in the mouse skeletal tissue.

In our studies we ensured that the force applied to the mouse cartilage and IVD explants is in line with published studies using murine tissue explants and cell cultures, most of which are within the MPa range and show a threshold effect (please see this meta-analysis by Natenstedt *et al* [1]).

This reviewer has raised a very interesting viewpoint on the possible differences in tissue mechanics of cartilage and IVD between mice and man. The force we used was calculated in Pa which is expressed as N/m². This means that the force scales with size. Although an

average human weighs ~2000 times more than the average mouse, the surface of the human joint and IVD over which the weight is distributed is proportionally larger than that of a mouse joint. Studies of mechanical properties of cartilage and IVDs from humans and various animal models have shown stiffness and Young's modulus within similar range regardless of animal size. For example, we and others have shown that in mouse knee cartilage and IVDs, Young's modulus was measured to be ~2 MPa on average [2] and 7.8 ± 1.5 MPa in another study [3]. While in human OA patients, the values were 4.46 ± 4.44 MPa in high weight bearing region and 9.81 ± 8.88 MPa in low weight bearing region [4]. In addition to similar mechanical properties the forces experienced by the tissues by articulation of the joints are also comparable. A study investigating forces *in vivo*, in live exposed but intact mouse knees, found that electrical activation of muscles resulted in pressure of 1.9 ± 0.2 MPa [5]. The IVD was also found to possess similar mechanical properties to humans. The average Young's modulus was 2-4 MPa and comparable to human motion segment (3-9 MPa) [6]. Moreover, a number of *ex vivo* studies found that compression of rodent skeletal tissues at a range of parameters similar to those used here in our study (0.2-1 MPa at 1 Hz) resulted in anabolic changes such as increased aggrecan and collagen II expression while loads over 1 MPa induced apoptosis [7–18]. Regardless of catabolic or anabolic effect, these studies do show that mouse tissues experience and react to comparable range of forces as human tissues.

Furthermore, daily variations in disc heights and tissue mechanics are believed to occur in quadruped mammals too, including mice. In fact, significant forces are required for quadrupeds to keep the fore-limbs and hind-limbs aligned to prevent spine sagging. Quadrupeds have evolved with buttressing mechanics around the spine like those of upright bipeds, with the forces created by the stabilizing musculature significantly compressing the intervertebral discs [19]. Quadruped mammals are known to experience atonic sleep during which they relax these stabilizing muscles, hence reducing the forces on the spine and joints [20].

We would also like to clarify that the *ex vivo* loading experiment was not designed to fully model the *in vivo* loading scheme, which as far as we are aware is technically challenging due to the size of the mouse skeletal system. Importantly our experiments were designed to illustrate the fact that direct mechanical loading, at a range similar to what has been shown by others as physiological/anabolic, can entrain the skeletal clocks in cartilage and IVD explants and in cell cultures without the confounding effects of humoral factors *in vivo*.

Restricted by the lack of direct measurements of mechanical loading parameters in mouse, and the limitation of the sensitivity of our loading equipment, we are not able to test the very low loading parameter (0.05 MPa) suggested by the reviewer. However, please do note that we had performed experiments using 0.2 MPa on cartilage and IVD explants and showed significant positive clock responses, namely a nearly 4-hour phase shift ($p < 0.001$) as shown in Fig. 1G (and in Fig.S3A, B of the original submission). In addition, we have tested a lower range of osmotic pressure increases (+20 and +50 mOsm) as a proxy and found that an increase as small as +20 mOsm was sufficient to induce a significant amplitude increase in primary chondrocytes and IVD cells, supporting our conclusions. This is now included as supplementary Fig. S4.

To acknowledge this reviewer's comments we have now added a caveat to the discussion (p.20, line 2-9): "In this study, we examined a range of physiologic and hyperphysiologic mechanical loading parameters; however, it is important to note that we are restricted by the current lack of direct measurements of the precise mechanical loading environment in mouse cartilage and IVD *in vivo*, as well as the limitation of the sensitivity of our loading equipment. Thus, we were not able to test a broader range of loading parameters. Future work is clearly warranted in this regard, including the development and testing of direct mechanical loading

systems in vivo to assess the minimal day/night differences required for entrainment of the cartilage and IVD clocks.”

2)RNA rhythm data of *Bmal1* and *Cry1* genes (Fig.1J) are fluctuating and not clear to demonstrate antiphasic gene expression rhythms between IVD cells that were subjected to compression cycles with antiphase. Why did not the authors examine more strongly rhythmic genes such as *Per2* (as shown in Fig.1A)?

We thank the reviewer for an excellent suggestion to better illustrate the anti-phasic nature of the clock gene oscillations following opposite loading cycles. We have performed cosinor curve fitting with 24-hour rhythmicity to the data, which hopefully is satisfactory to address this issue. Please see new Fig. 1J.

In our hands, the *Per2* gene rhythm in rat immortalised IVD cells is not as robust as *Bmal1* or *Cry2*, hence why we did not include the *Per2* data. However, this does not affect our conclusions.

3) The experimental procedures for measurement of cellular rhythms are not clearly stated; “PER2::Luc IVD explants were cultured under static osmotic conditions of 230-730 mOsm for 3 days before clock synchronisation and recording” (p.7, line 16-). Did they synchronize cells by DEX “pulse” treatment? Which culture medium was used after synchronization? Did they use a medium with normal osmolarity after synchronization? If so, these data do not demonstrate that “the circadian period in these skeletal tissues is osmolarity-compensated (Fig.2A)”. Most probably, the osmolarity has an obvious effect on the cellular rhythms as demonstrated in Fig.2B, and the rhythms are “not compensated”. Rather, the Fig.1A data indicate that (potential) changes in circadian period in hyperosmotic medium is “reversible” when returned back to the normal medium.

We apologise for not making this clear in the manuscript. Explants were cultured before and after Dex in media with the stated osmolarity. They did not return to normal osmolarity in recording media. Therefore, the period of the cartilage and IVD circadian clocks is “osmolarity-compensated” because it is relatively insensitive to the drastic osmolarity differences between the culture conditions. Dex treatment was a pulse of 1 hour.

We have modified the sentence to make it clearer by adding “then resynchronised with dexamethasone and recorded in media at the adapted osmolarity as indicated.” at p.7 line 26 and 8, line 1.

Similarly, procedures for Fig.2B is unclear: “..and allowed to adapt for 3 days before synchronisation and recording.” (p.8, line 1) Did they synchronize cells by DEX “pulse” ?

We have now provided a detailed description of the experiments in experimental procedures in the “Tissue explant culture, mechanical loading and osmotic challenge” section.

In Fig.2B, the waveform in control in black is invisible, and hence the blue and red lines cannot be compared with the control.

A Dex pulse of 1 hour was applied at the beginning of the experiments. There was no further Dex synchronisation during the experiment, only an osmolarity media swap.

We have made the waveform of the control in Figure 2B more visible now, thus allowing the test regimes (blue and red lines) to be compared to the control.

4) It is not clear to me how the results of single-cell imaging in Fig.2C, D, E are related to the other part of the manuscript. In addition, I cannot find detailed explanation for preparation of the heatmap in Fig.1E and how the bottom bar graph was prepared.

This analysis was designed to assess whether hyper-osmolarity synchronises clocks at cell population level or increases clock gene expression in individual cells. It also allowed us to evaluate the percentage of cells that responded, which we would not be able to ascertain using PER2::Luc recording. These experiments also further consolidated our conclusions that an increase in osmotic pressure resets clocks by enhancing rhythmicity of PER2 nuclear accumulation. We have added additional details about the rationale for the single cell imaging experiments (p.8, line 19-20): “To assess whether hyper-osmolarity synchronises clocks at an individual cell level and to evaluate the percentage of cells that respond, single-cell fluorescence imaging of PER2::Venus mouse primary chondrocytes were performed.”

More details are now provided for the heatmap generation and cell tracking in experimental procedures “PER2-Venus live cell imaging” section and an explanation for the bar graph has been added to the Fig 2E legend.

5) Phase transition curve (Fig.S2D) should be plotted in CT for both the pulse time and new phase. Otherwise, the slope is not 1 for type 1 resetting.

Thank you to the reviewer for raising this point. We have replotted the PRC and PTC using CT. Please see the new Fig. S2C, D.

6) In Fig. 2, the phase-shift is very large (almost up to 10 hours) in +400 experiment. Does this stimulus still show a type-1 resetting? or, does the type 1 phase-shift change to type-0 resetting in response to stronger stimuli?

The reviewer has raised an excellent point. As has been predicted by Winfree, Type-1 resetting and Type-0 resetting to light have been demonstrated in humans, depending on stimulus strength [21–23].

Following the reviewer’s suggestion, we performed additional experiments with the aim to explore the PRC at +400 mOsm. Interestingly, we found at all 4 circadian phases tested, the

clock rhythmicity is severely disrupted. When these data were normalized, we could still show a large phase delay, as we had showed previously. However, we do not feel it appropriate to quantify the phase shifts when the rhythm was severely suppressed. These results suggest that +400 mOsm may have exceeded the physiological range of osmotic changes. Indeed, our earlier data also indicated +300 mOsm as optimal for the induction of clock rhythm.

We have modified the sentence at p.9, line 10-13 which now reads: “Significant amplitude effect was detectable at an increase as small as +20 mOsm, and maximal amplitude induction was observed with +300 mOsm condition, suggesting higher osmotic challenge may exceed the physiological range (Fig. 2H, Fig. S4).”

7) Why do the authors demonstrate the results with a different reporter Cry1-luc mice in Fig.S3? Apparently, the data are slightly different from those obtained in Per2-luc mice, but no explanation is given for the experiments in Fig.S3.

We performed these experiments to show that the molecular circadian clocks in cartilage and IVDs are entrainable to loading and osmolarity, irrespective of the clock reporters used. The Cry1-Luc is a promoter reporter and PER2::Luc is a fusion protein. The Cry1-Luc looks slightly different (mainly due to a drift in baseline signals) but still showed increased amplitude and phase shifts after loading and osmotic stress in a stimulus magnitude-dependent manner. We have now added quantification of the Cry1-Luc data by loading (new Fig.S5).

We have added the rationale to the text (p.9, line 13-15): “To further validate the responses of molecular circadian clocks to loading and osmolarity, we used cartilage and IVD tissue explants from a different clock reporter mouse model, the Cry1-Luc which is a promoter reporter²⁷ as opposed to the PER2::Luc fusion protein reporter²⁸.”

8) In experiments in Fig.3, they compared the effect of one daily cycle of +100 mOsm treatment with two daily cycles of +200 mOsm. Therefore, it is not concluded whether the number of cycles was effective or the strength of the osmolarity was effective (or both). It is required to revise the following statement; .. this effect was clearly dependent on the extent of osmotic change and number of cycles applied. (p.10, line 16)

We have revised the statement (p.11, line 16-17). It now reads: "this effect seems to be dependent on the extent of osmotic change, or the number of cycles applied, or both".

9) The authors concluded that cellular rhythms from U2OS in culture was not sensitive to application of osmolarity change, but the experiment was performed only at a single time point of the day (Fig.S4B). Because the osmolarity-sensitivity was not observed at a certain time of the day (at Per2 trough) even in skeletal tissues (Fig.2F), it is not easy to conclude that "the osmolarity-entrainment is clearly cell type dependent" from the current experimental data (as stated in p.12, line 12).

We have revised the statement (p.13, line 10-11). It now reads: "As such, the osmolarity-entrainment of circadian clock is likely cell type dependent and could indicate cellular adaptation to their local niche."

10) In the principal component analysis of the RNA seq data, the three replicates of control Sedentary group are broadly scattered as compared with those of the treadmill Running group in both the cartilage and IVD samples (Fig.S5A). What is the possible reason for this.

The PCA analysis reduces the dimensionality of the data while retaining most of the variation in the dataset. As such, we can only conclude that there is less variability in global gene expression profiles among the treadmill running samples than among the sedentary group. This could mean that treadmill running uniformly induces (or suppresses) gene expression changes while in the sedentary group these genes are at different levels. However, it is important to note that, this is a speculation at best.

11) The pharmacological experiments in Fig.5 revealed that PLD2-AKT-mTORC2 is involved in osmolarity-sensitive clock regulation. Interestingly, phosphorylation level of GSK3beta is affected by the osmolarity change, but inhibition of GSK3beta by lithium treatment did not weaken the osmolarity-sensitive amplification, but it rather strengthened the rhythms (Fig.5E). Therefore, change in GSK3beta phosphorylation upon osmolarity change may be irrelevant to the clock regulation. Please clarify the phrase; "as well as phosphorylation of GSK3beta after hyperosmotic stress (Fig 5K)". (p. 12. line 9)

We apologise for not making it clear in the manuscript. We observed increased phosphorylation of GSK3b at Ser9 and Ser389 after hyperosmotic shock. The phosphorylation events at these two sites have inhibitory effect on GSK3b activity [24,25]. Therefore, treatment with lithium, a GSK3b inhibitor, should have similar effect on PER2 rhythm as increased phosphorylation of GSK3b at Ser9 and Ser389 which is exactly what we observed.

12) Literature search revealed a recent paper reporting that mTORC2-AKT is

responsible for resetting of fibroblast clock triggered by osmotic stress (doi: 10.1089/ars.2021.0059.) Together, it appears that the osmolarity-sensitive regulation of cellular clock is common to a variety of cells, and this idea is diverged from the authors' speculation that the signaling pathway is cell-type-dependent (Comment 9). Information about the expression levels of PLD2, AKT or mTORC2 among tissues/cells should help discussion about the issue.

Thank you to the reviewer for pointing out this relevant paper. We are pleased to know that the hyperosmolarity-induced clock synchronisation pathways we found in skeletal system agree with the clock resetting (phase-shifting) pathways in fibroblasts as shown elegantly by Yoshitane et al 2022, including mTOR and AKT. We agree with this reviewer that a variety of cells may show a clock response to osmolarity challenge. However, for cell types that do not typically experience a drastic change in osmolarity between day and night, they may respond to such osmotic challenges by a shift in rhythm (resetting), or a dampening of clock amplitude (as we observed in U2OS and keratinocytes), instead of clock entrainment. In other words, for those cell types, daily loading pattern and associated osmotic changes are less likely to be a physiological zeitgeber/time cue.

Where appropriate we have toned down our statements as we have not fully characterised a number of different cell types and tissues by treatment across a full circadian cycle. As our response to point 9), we modified it to "As such, the osmolarity-entrainment of circadian clock is likely cell type dependent and could indicate cellular adaptation to their local niche."

And in the discussion, we have revised this sentence (p.21, line 18-21): "However, in their *in vitro* studies cells exhibited type 0 resetting and reacted to both hyper and hypo osmotic conditions while our tissue explants showed type 1 resetting at +200 mOsm challenge and reacted to hyperosmolarity only, suggesting cell type and tissue specific clock responses". We have also included this new reference in the discussion (p.21, line 11-13), by adding: "Supporting our findings of underlying pathways, mTOR-AKT signaling has recently been implicated in cellular clock phase shifting triggered by hyper-osmotic stress in cultured fibroblasts⁴⁵."

Minor points;

IPA should be spelled out in p.13, line 17. Then, Ingenuity Pathway Analysis (p.19, line 10) can be abbreviated.

This has been changed.

Reviewer #3 (Remarks to the Author):

These data support the central premise that mechanical force might be an entrainment factor for cartilage, as compelling data indicate augmentation of different elements of rhythmicity and clock genes in response to force and changes in osmolarity.

The use of both in vivo and ex vivo (uncoupling systemic signals) is a real strength here.

We thank this reviewer for their kind support of our work.

Minor: It might be helpful for the non-expert reader to be reminded why the Per2 reporter is such a relevant read-out here.

We have now included a sentence in methods (p32, line 1-7) about the PER2 and Cry1 reporter mouse models.

“To investigate tissue level responses to treadmill exercise *in vivo* and mechanical loading/osmotic challenge *ex vivo*, PER2::Luc/PER2-Venus and *Cry1*-Luc reporter mouse lines were chosen because: 1) these well-established clock gene reporters represent the negative arm of the molecular clock which has been shown to be highly rhythmic and is a faithful reflection of the internal tissue clocks; and 2) they represent different levels of clock gene regulation, with PER2::Luc and PER2-Venus being protein fusions (protein level reporter), while *Cry1*-Luc being promoter driven (transcriptional reporter)”.

Equally, the data strongly support that acute hypertonic shock augments rhythmicity in a manner similar to effects of loading.

Minor: It might again be useful for non-expert to point out that cartilage has a high fixed charge density supporting such fluctuations with loading.

We have added a sentence in the introduction (p.3, line 23, 24 and p.4 line 1-3) about the high fixed charge density of cartilage and IVD tissue: “Tissues such as cartilage and IVD have high negative charges attached to the proteoglycan-rich matrix in each tissue unit/volume, which significantly affects tissue mechanical properties (including the compressive modulus) and swelling pressure. As such, cartilage and IVD tissues show profound fluctuations in their osmotic environment upon daily mechanical loading^{14,15}.”

It is slightly hard for the reader to make direct comparisons between responses to loading and hyperosmotic shock (and thus access relative “phenocopy”) but nevertheless the effects do appear similar in nature. Perhaps some side-by side analysis in supplementaries would help. The authors do clearly state differences (for example in amplitude effects and later in molecular mechanisms supporting – e.g ERK signalling).

We have included a new supplementary figure (Fig. S4) summarising the similarities and differences in tissue responses (amplitude, phase, and dose responses) to loading vs. osmolarity. We hope that this helps the reader to make direct comparisons between loading and hyperosmotic shock as it clearly shows a similar magnitude and direction of changes between the two stimuli.

The similarities and physiologically relevant association of force and osmolarity in cartilage rationalise dissecting the effects using agnostic sequencing to find a point of convergence. The identification and detailed dissection of the shared molecular mechanism is a final strength of the study.

This reviewer congratulates the researchers on successfully finding this after chasing multiple avenues. A central challenge here was always going to be supporting the notion that the effects of mechanical force on cycling are mediated by a tonic flux given inhibiting the later during the application of load is technically challenging. The hunt for a shared 'sensor' i guess continues.

We thank the reviewer for their comment about the possibility of a “shared sensor (s)” and we will continue to hunt for a shared sensor (s), especially those on the cell membrane, in future project!

This is certainly not suggested as a necessary experiment, but do the authors think, or perhaps have any data that speaks to the idea, that old or diseased cartilage, with degraded negatively charged proteoglycan, might exhibit a differential response to loading due to an altered (diminished?) capacity for an osmotic flux? Perhaps the results, as for osmolarity shown in the manuscript, will be the same- as the changes are at the level of cells machinery to sense/transduce environmental changes. It will of course be hard/impossible to uncouple the confounding effects of other changes in the (old/diseased) tissue. Similarly, perhaps an approach using “engineered” tissue/scaffolds with different charge properties, seeded with PER2 reporting chondrocytes might illuminate strengthen further the link between force and osmotic changes.

Thank you to the reviewer for an excellent suggestion for future work! We totally agree that a tissue engineering approach as you kindly suggested will allow us to tease out the contributions of the matrix charge properties to the effect of loading on clocks.

We have previously shown that cartilage/IVD clocks dampen with ageing in mouse model. We think the dampening of circadian rhythms in these tissues are likely multi-factorial and may involve both cell-intrinsic (e.g. senescence, age-related oxidative stress, dysregulated expression of cell surface sensors/signaling mechanisms, etc) and systemic changes (inefficient time cues due to a sedentary lifestyle during ageing).

We have not performed loading experiments on aged/diseased tissues. As this reviewer rightly pointed out, these experiments are likely confounded by secondary factors that are not related directly to loading nor osmolarity changes.

This work significantly contextualises much of the earlier work by this group in this arena and the thoughtful discussion frames some of the future avenues that might be explored to translate these cell and tissue phenomena to tissue health context. As written, the work certainly supports the conclusions drawn and I see no flaws in design or interpretation prohibiting publication. The group have previously shown the strengths of the methodology used, which is highly appropriate and is all the stronger for the multiple tissue types and primary cell types used. Description of methods relies heavily on their former description when used previously but as written the reader can follow what is being done.

Thank you so much for your kind words!

References cited in the responses to the reviewers:

- [1] J. Natenstedt, A.C. Kok, J. Dankelman, G.J. Tuijthof, What quantitative mechanical loading stimulates in vitro cultivation best?, *J. Exp. Orthop.* 2 (2015) 15. <https://doi.org/10.1186/s40634-015-0029-x>.
- [2] L. Cao, I. Youn, F. Guilak, L.A. Setton, Compressive Properties of Mouse Articular Cartilage Determined in a Novel Micro-Indentation Test Method and Biphasic Finite

- Element Model, *J. Biomech. Eng.* 128 (2006) 766–771.
<https://doi.org/10.1115/1.2246237>.
- [3] A. Kotelsky, A. Elahi, C. Nejat Yigit, A. Proctor, S. Mannava, C. Pröschel, W. Lee, Effect of knee joint loading on chondrocyte mechano-vulnerability and severity of post-traumatic osteoarthritis induced by ACL-injury in mice, *Osteoarthr. Cartil. Open.* 4 (2022) 100227. <https://doi.org/10.1016/j.ocarto.2021.100227>.
- [4] A.A. Mieloch, M. Richter, T. Trzeciak, M. Giersig, J.D. Rybka, Osteoarthritis Severely Decreases the Elasticity and Hardness of Knee Joint Cartilage: A Nanoindentation Study, *J. Clin. Med.* 8 (2019) 1865. <https://doi.org/10.3390/jcm8111865>.
- [5] Z. Abusara, R. Seerattan, A. Leumann, R. Thompson, W. Herzog, A novel method for determining articular cartilage chondrocyte mechanics in vivo, *J. Biomech.* 44 (2011) 930–934. <https://doi.org/10.1016/j.jbiomech.2010.11.031>.
- [6] D.M. Elliott, J.J. Sarver, Young Investigator Award Winner: Validation of the Mouse and Rat Disc as Mechanical Models of the Human Lumbar Disc, *Spine.* 29 (2004) 713. <https://doi.org/10.1097/01.BRS.0000116982.19331.EA>.
- [7] A.J.L. Walsh, J.C. Lotz, Biological response of the intervertebral disc to dynamic loading, *J. Biomech.* 37 (2004) 329–337. [https://doi.org/10.1016/S0021-9290\(03\)00290-2](https://doi.org/10.1016/S0021-9290(03)00290-2).
- [8] C. Bougault, M. Gosset, X. Houard, C. Salvat, L. Godmann, T. Pap, C. Jacques, F. Berenbaum, Stress-induced cartilage degradation does not depend on the NLRP3 inflammasome in human osteoarthritis and mouse models, *Arthritis Rheum.* 64 (2012) 3972–3981. <https://doi.org/10.1002/art.34678>.
- [9] D.-L. Wang, S.-D. Jiang, L.-Y. Dai, Biologic response of the intervertebral disc to static and dynamic compression in vitro, *Spine.* 32 (2007) 2521–2528. <https://doi.org/10.1097/BRS.0b013e318158cb61>.
- [10] J.J. MacLean, C.R. Lee, M. Alini, J.C. Iatridis, Anabolic and catabolic mRNA levels of the intervertebral disc vary with the magnitude and frequency of in vivo dynamic compression, *J. Orthop. Res.* 22 (2004) 1193–1200. <https://doi.org/10.1016/j.orthres.2004.04.004>.
- [11] J.J. Maclean, C.R. Lee, M. Alini, J.C. Iatridis, The effects of short-term load duration on anabolic and catabolic gene expression in the rat tail intervertebral disc, *J. Orthop. Res.* 23 (2005) 1120–1127. <https://doi.org/10.1016/j.orthres.2005.01.020>.
- [12] J.J. MacLean, C.R. Lee, S. Grad, K. Ito, M. Alini, J.C. Iatridis, Effects of Immobilization and Dynamic Compression on Intervertebral Disc Cell Gene Expression In Vivo, *Spine.* 28 (2003) 973. <https://doi.org/10.1097/01.BRS.0000061985.15849.A9>.
- [13] Y. Xing, P. Zhang, Y. Zhang, L. Holzer, L. Xiao, Y. He, R. Majumdar, J. Huo, X. Yu, M.K. Ramasubramanian, L. Jin, Y. Wang, X. Li, J. Oberholzer, A multi-throughput mechanical loading system for mouse intervertebral disc, *J. Mech. Behav. Biomed. Mater.* 105 (2020) 103636. <https://doi.org/10.1016/j.jmbbm.2020.103636>.
- [14] J.C. Lotz, J.R. Chin, Intervertebral disc cell death is dependent on the magnitude and duration of spinal loading, *Spine.* 25 (2000) 1477–1483. <https://doi.org/10.1097/00007632-200006150-00005>.
- [15] J. Li, Y. Ma, Y. Jiao, L. Xu, Y. Luo, J. Zheng, X. Zhang, Z. Chen, Intervertebral Disc Degeneration and Low Back Pain Depends on Duration and Magnitude of Axial Compression, *Oxid. Med. Cell. Longev.* 2022 (2022) 1045999. <https://doi.org/10.1155/2022/1045999>.
- [16] K. Ariga, K. Yonenobu, T. Nakase, N. Hosono, S. Okuda, W. Meng, Y. Tamura, H. Yoshikawa, Mechanical Stress-Induced Apoptosis of Endplate Chondrocytes in Organ-Cultured Mouse Intervertebral Discs: An Ex Vivo Study, *Spine.* 28 (2003) 1528. <https://doi.org/10.1097/01.BRS.0000076915.55939.E3>.
- [17] C. Bougault, S. Priam, X. Houard, A. Pigenet, L. Sudre, R.J. Lories, C. Jacques, F. Berenbaum, Protective role of frizzled-related protein B on matrix metalloproteinase induction in mouse chondrocytes, *Arthritis Res. Ther.* 16 (2014) R137. <https://doi.org/10.1186/ar4599>.

- [18] E. Pecchi, S. Priam, M. Gosset, A. Pigenet, L. Sudre, M.-C. Laiguillon, F. Berenbaum, X. Houard, Induction of nerve growth factor expression and release by mechanical and inflammatory stimuli in chondrocytes: possible involvement in osteoarthritis pain, *Arthritis Res. Ther.* 16 (2014) R16. <https://doi.org/10.1186/ar4443>.
- [19] T.H. Smit, The use of a quadruped as an in vivo model for the study of the spine – biomechanical considerations, *Eur. Spine J.* 11 (2002) 137–144. <https://doi.org/10.1007/s005860100346>.
- [20] J.C. Fryer, Is a purpose of REM sleep atonia to help regenerate intervertebral disc volumetric loss?, *J. Circadian Rhythms.* 7 (2009) 1. <https://doi.org/10.1186/1740-3391-7-1>.
- [21] C.A. Czeisler, R.E. Kronauer, J.S. Allan, J.F. Duffy, M.E. Jewett, E.N. Brown, J.M. Ronda, Bright Light Induction of Strong (Type 0) Resetting of the Human Circadian Pacemaker, *Science.* 244 (1989) 1328–1333. <https://doi.org/10.1126/science.2734611>.
- [22] S.B.S. Khalsa, M.E. Jewett, C. Cajochen, C.A. Czeisler, A phase response curve to single bright light pulses in human subjects, *J. Physiol.* 549 (2003) 945–952. <https://doi.org/10.1113/jphysiol.2003.040477>.
- [23] M.E. Jewett, R.E. Kronauer, C.A. Czeisler, Light-induced suppression of endogenous circadian amplitude in humans, *Nature.* 350 (1991) 59–62. <https://doi.org/10.1038/350059a0>.
- [24] X. Fang, S.X. Yu, Y. Lu, R.C. Bast, J.R. Woodgett, G.B. Mills, Phosphorylation and inactivation of glycogen synthase kinase 3 by protein kinase A, *Proc. Natl. Acad. Sci.* 97 (2000) 11960–11965. <https://doi.org/10.1073/pnas.220413597>.
- [25] T.M. Thornton, G. Pedraza-Alva, B. Deng, C.D. Wood, A. Aronshtam, J.L. Clements, G. Sabio, R.J. Davis, D.E. Matthews, B. Doble, M. Rincon, Phosphorylation by p38 MAPK as an alternative pathway for GSK3beta inactivation, *Science.* 320 (2008) 667–670. <https://doi.org/10.1126/science.1156037>.
- [26] A.W. Palmer, R.E. Guldberg, M.E. Levenston, Analysis of cartilage matrix fixed charge density and three-dimensional morphology via contrast-enhanced microcomputed tomography, *Proc. Natl. Acad. Sci. U. S. A.* 103 (2006) 19255–19260. <https://doi.org/10.1073/pnas.0606406103>.
- [27] Y. Wu, S.E. Cisewski, Y. Sun, B.J. Damon, B.L. Sachs, V.D. Pellegrini, E.H. Slate, H. Yao, Quantifying Baseline Fixed Charge Density in Healthy Human Cartilage Endplate: A Two-point Electrical Conductivity Method, *Spine.* 42 (2017) E1002–E1009. <https://doi.org/10.1097/BRS.0000000000002061>.

REVIEWERS' COMMENTS

Reviewer #2 (Remarks to the Author):

I think that the authors responded faithfully and honestly to each of my review comments given to their originally submitted manuscript. I have no further criticism on the revised manuscript, and now I feel happy to have participated in the reviewing process of this article.

Reviewer #3 (Remarks to the Author):

The authors have addressed the concerns and suggestions of this reviewer.